# Fast and flexible sequence induction in spiking neural networks via rapid excitability changes

**Rich Pang[1,2,3]\*, Adrienne L Fairhall[2,3]**

[1]Neuroscience Graduate Program, University of Washington, Seattle, United States; [2]Department of Physiology and Biophysics, University of Washington, Seattle, United States; [3]Computational Neuroscience Center, University of Washington, Seattle, United States

**Abstract** Cognitive flexibility likely depends on modulation of the dynamics underlying how biological neural networks process information. While dynamics can be reshaped by gradually modifying connectivity, less is known about mechanisms operating on faster timescales. A compelling entrypoint to this problem is the observation that exploratory behaviors can rapidly cause selective hippocampal sequences to 'replay' during rest. Using a spiking network model, we asked whether simplified replay could arise from three biological components: fixed recurrent connectivity; stochastic 'gating' inputs; and rapid gating input scaling via long-term potentiation of intrinsic excitability (LTP-IE). Indeed, these enabled both forward and reverse replay of recent sensorimotor-evoked sequences, despite unchanged recurrent weights. LTP-IE 'tags' specific neurons with increased spiking probability under gating input, and ordering is reconstructed from recurrent connectivity. We further show how LTP-IE can implement temporary stimulus-response mappings. This elucidates a novel combination of mechanisms that might play a role in rapid cognitive flexibility.

DOI: https://doi.org/10.7554/eLife.44324.001

**\*For correspondence:**
rpang@uw.edu

## Introduction

We can rapidly and flexibly adapt how we process incoming information from the environment, a mental faculty known as cognitive flexibility. For example, after being instructed to raise our left hand when one word is heard and our right hand when another is heard, we perform the task with little error. How brain networks quickly induce novel stimulus-response mappings such as this into their underlying neural dynamics, however, remains mysterious. In particular, while extensive prior work has elucidated how stimulus-response mappings might be implemented biologically, the mechanisms for inducing these mappings typically require slow, gradual modifications to network structure, for example by incrementally training connection weights to minimize errors between correct and predicted responses (*Williams and Zipser, 1989*; *Sussillo and Abbott, 2009*; *Laje and Buonomano, 2013*; *Rajan et al., 2016*; *Nicola and Clopath, 2017*) or by allowing local plasticity to reshape network dynamics in response to internal activity (*Song and Abbott, 2001*; *Fiete et al., 2010*; *Gilson et al., 2010*; *Lee and Buonomano, 2012*; *Klampfl and Maass, 2013*; *Rezende et al., 2011*; *Diehl and Cook, 2015*). Biologically observed spike-timing-dependent plasticity (STDP) mechanisms, however, typically increase post-synaptic potentials (PSPs) by at most a few percent (*Markram et al., 1997*; *Bi and Poo, 1998*; *Sjöström et al., 2001*; *Wang et al., 2005*; *Caporale and Dan, 2008*). Furthermore, due to the precise timing requirements of canonical plasticity mechanisms (e.g. in STDP spike pairs must occur in the correct order within tens of milliseconds) and low firing rates of cortex and hippocampus (typically less than a few tens of Hz [*Griffith and Horn, 1966*;

*Koulakov et al., 2009*; *Mizuseki and Buzsáki, 2013*]), STDP-triggering spike patterns may occur relatively rarely, especially given the asynchronous nature of cortical firing patterns in awake animals (*Renart et al., 2010*). As a result, computational models for shaping dynamics that modify network structures via synaptic plasticity typically rely on at least dozens of learning trials over extended time periods (*Song and Abbott, 2001*; *Masquelier et al., 2008*; *Klampfl and Maass, 2013*), challenging their suitability for rapid reconfiguration of network dynamics. Computationally, modifying connections to change network function might also interfere with long-term memories or computations already stored in the network's existing connectivity patterns, leading, for instance, to catastrophic forgetting (*McCloskey and Cohen, 1989*). Consequently, it is unclear (1) how biologically observed plasticity mechanisms could reshape network function over the timescales of seconds required for rapid cognitive flexibility, and (2) how such restructuring of synaptic connectivity could occur over the short term without degrading existing long-term memories.

One feature of neural dynamics potentially reflecting processes of cognitive flexibility is stereotyped sequential firing patterns (*Hahnloser et al., 2002*; *Ikegaya et al., 2004*; *Luczak et al., 2007*; *Pastalkova et al., 2008*; *Davidson et al., 2009*; *Crowe et al., 2010*; *Harvey et al., 2012*). Functionally, firing sequences are thought to be involved in various cognitive processes, from short-term memory (*Davidson et al., 2009*; *Crowe et al., 2010*) to decision-making (*Harvey et al., 2012*). More generally, one can imagine sequential activity as reflecting information propagating from one subnetwork to another, for example stimulus S evoking a cascade of activity that eventually triggers motor output M. A compelling empirical example of memory-related firing sequences that arise in a neural network almost immediately after a sensorimotor event, apparently without requiring repeated experience or long-term learning, occurs in awake hippocampal 'replay'. Here, sequences of spikes in hippocampal regions CA1 and CA3 originally evoked by a rodent traversing its environment along a specific trajectory subsequently replay when the rodent pauses to rest (*Foster and Wilson, 2006*; *Davidson et al., 2009*; *Gupta et al., 2010*; *Carr et al., 2011*). Such replay events occur at compressed timescales relative to the original trajectory and often in reverse order. Replay has also been observed in primate cortical area V4, where firing sequences evoked by a short movie were immediately reactivated by a cue indicating the movie was about to start again, but without showing the movie (*Eagleman and Dragoi, 2012*). The functional role of replay has been implicated in memory, planning, and learning (*Ego-Stengel and Wilson, 2010*; *Carr et al., 2011*; *Eagleman and Dragoi, 2012*; *Jadhav et al., 2012*; *Ólafsdóttir et al., 2018*), but little is known about the mechanisms enabling the underlying sequential activity to be induced in the network dynamics in the first place. Elucidating biological mechanisms for rapidly inducing sequential firing patterns in network dynamics may not only illuminate the processes enabling replay but may also shed light on principles for fast and flexible reconfiguration of computations and information flow in neural networks more generally.

An intriguing fast-acting cellular mechanism whose role in shaping network dynamics has not been investigated is the rapid, outsize, and activity-dependent modulation of cortical inputs onto pyramidal cells (PCs) in hippocampal region CA3 (*Hyun et al., 2013*; *Hyun et al., 2015*; *Rebola et al., 2017*). Specifically, following a 1–2 s train of 20 action potentials in a given CA3 PC, excitatory postsynaptic potentials (EPSPs) from medial entorhinal cortex (MEC) more than doubled in magnitude within eight seconds (the first time point in the experiment). Thought to arise through inactivation of $K^+$-channels colocalized with MEC projections onto CA3 PC dendrites and deemed 'long-term potentiation of intrinsic excitability' (LTP-IE), potentiation occurred regardless of whether the CA3 PC spikes were evoked via current injection or by upstream physiological inputs, indicating its heterosynaptic nature, since only MEC EPSPs, and not others, exhibited potentiation (*Hyun et al., 2013*; *Hyun et al., 2015*). Notably, the 10–20 Hz spike rate required for potentiation matches the range of in vivo spike rates in hippocampal 'place cells' when rodents pass through specific locations (*Moser et al., 2008*; *Mizuseki et al., 2012*), suggesting LTP-IE may occur in natural contexts. Furthermore, EPSP potentiation persisted throughout the multi-minute course of the experiment (*Hyun et al., 2015*), suggesting that in addition to fast onset, the modulation could extend significantly into the future. Although inducing sequential activation patterns in neural networks is typically associated with *homosynaptic* plasticity (e.g. in STDP a postsynaptic following a presynaptic spike strengthens the activated synapse, thereby incrementally increasing the probability of subsequent presynaptic spikes triggering postsynaptic spikes), the rapidity, strength, and duration of this heterosynaptic potentiation mechanism suggest it might significantly modulate network

dynamics in natural conditions, warranting further investigation within the context of neural sequences. Moreover, while this mechanism has so far been observed in hippocampus only, an intriguing possibility is that functionally similar mechanisms exist in cortex also, enabling rapid effective changes in excitability of recently active cells, with potentially similar computational consequences (see Discussion).

To explore this idea we first develop a computational model for the effect of LTP-IE on EPSPs from upstream inputs as a function of their spiking responses to physiological inputs. Next, we demonstrate how LTP-IE combined with recurrent PC connectivity in a spiking network can yield spike sequences reflecting recent sensorimotor sequences that replay in both forward and reverse and which are gated by an upstream gating signal. We subsequently identify parameter regimes allowing and prohibiting LTP-IE-based sequence propagation and examine the effect of specific parameters on replay frequency and propagation speed. We next show how LTP-IE-based sequences can be used to induce temporary stimulus-response mappings in an otherwise untrained recurrent network. Finally, using a reduced model we give proof-of-concept demonstrations of how LTP-IE might support more general computations. We discuss implications for cognitive flexibility and rapid memory storage that does not require modification of recurrent network weights.

## Results

To investigate its consequences on neuronal spiking dynamics we implemented LTP-IE in a network of leaky integrate-and-fire (LIF) neurons (*Gerstner, 2014*). Neurons received three types of inputs: sensorimotor inputs S, carrying tuned 'external' information; recurrent input R from other cells in the network, comprising excitation and inhibition (developed further, Figure 2); and 'gating' inputs G, assumed to be random, with a homogeneous rate across the network but independent to each cell. We provide an overview and intuition of the model before elaborating our results.

First, we demonstrate our implementation of LTP-IE in spiking neurons. We apply LTP-IE to a neuron, potentiating its G inputs, if the neuron fires within a physiological range (~10–20 Hz in our model) for 1 s in response to input S. Thus, recently active cells get 'tagged' by LTP-IE, so that they exhibit augmented EPSPs in response to G inputs. In the presence of inputs G, recently active cells will therefore show larger positive membrane voltage deflections, on average, because their EPSPs from G are larger, and they will sit closer to spiking threshold. Unless otherwise specified, in all simulations that follow we assume the continuous presence of inputs G, as LTP-IE would not affect membrane voltages without them.

Second, in analogy to hippocampal 'place cells' (*Moser et al., 2008*; *Mizuseki et al., 2012*) we consider the case where different excitatory pyramidal cells (PCs) maximally fire (up 20 Hz) when a simulated animal is in different positions in the environment. When the simulated animal moves along a trajectory, LTP-IE thus tags the cells with place fields along the trajectory, storing the trajectory memory as a set of LTP-IE-tagged neurons with specific position tunings; order and timing information, however, are not stored by LTP-IE. When G inputs are turned on post-navigation (representing an awake resting state), cells activated by the original trajectory thus sit closer to spiking threshold than cells not activated by the trajectory.

Third, inspired by excitatory recurrence in CA3 (*Lisman, 1999*) we assume recurrent excitatory connections in the network, which allows activity to propagate through the network. We assume that cell pairs with more similar tuning (i.e. with nearby place fields) have stronger connections (*Lisman, 1999*; *Káli and Dayan (2000)*; *Giusti et al., 2015*). Thus, activity from cell A is more likely to propagate to cell B if (1) cell B has been LTP-IE-tagged after being activated by the trajectory, and (2) cell B has a nearby place field to cell A. Consequently, given the right model parameters, activity that begins at one point in the network (e.g. in cells tuned to the animal's resting position) should propagate among LTP-IE-tagged cells, in an order reflective of the original trajectory.

Finally, we include a population of inhibitory cells (INH) one-tenth the size of the excitatory PC population and randomly recurrently connected with them. Inhibition limits the number of active PCs and prevents activity from propagating in more than one direction along the LTP-IE-defined trajectory. We discuss additional model components such as LTP-IE extinction and higher-order structure in G inputs in the Discussion.

Note that our model accords two meanings of 'excitability' to the LTP-IE acronym. As coined by *Hyun et al. (2013)* and *Hyun et al. (2015)*, LTP-IE's 'excitability' originally refers to the augmented

EPSPs of G inputs (MEC inputs, in *Hyun et al., 2013*; *Hyun et al., 2015*) arriving to distal dendrites. Here, however, 'excitability' also refers to the increased membrane potential of LTP-IE-tagged cells (in the presence of G inputs), which makes them more likely to spike in response to recurrent inputs.

## LTP-IE increases membrane voltages and spike rates under random gating inputs

We first simulate the expected effects of LTP-IE on neuronal membrane voltages and spike rates. To model LTP-IE, when a cell spikes sufficiently fast (~10–20 Hz) over 1 s, we scale that cell's G inputs by a factor $\sigma(r)$ that depends on the rate $r$ at which it fired during that 1 s. For such cells, future G inputs will thus yield augmented EPSPs (*Figure 1A,B*). Consequently, cells that have recently emitted several spikes in quick succession end up with high LTP-IE levels, which causes them to sit at higher average voltages given a steady random stream of G inputs (*Figure 1C–E*); additionally, cells with higher LTP-IE exhibit increased variability in their membrane voltages (*Figure 1D,E*). (Note: we chose to model the LTP-IE activation function [*Figure 1B*] as a sigmoid based on the saturation-like relationship between PC spike count and the membrane conductance change measured in *Hyun et al. (2015)* [Figure 3H], but evidence also exists for an optimal firing rate between 10 and 20 Hz [*Hyun et al., 2013*]; modeling LTP-IE activation with such an optimum should not affect our results, however, as our goal was simply to apply LTP-IE to PCs firing at physiological rates.)

Due to their increased membrane voltage, cells with high LTP-IE also exhibit higher spiking probability, both spontaneously (*Figure 1—figure supplement 1*) and in response to depolarizing input currents (*Figure 1F,G*), although even with high LTP-IE, current-evoked spiking is not assured since substantial variability remains (*Figure 1F,G*). Nonetheless, LTP-IE can increase a model neuron's probability of transforming an input current into an output spike from near zero to approximately 0.5, when the PC→G synaptic weight $w^{PC,G}$ takes a moderate value (*Figure 1G*). (For overly weak $w^{PC,G}$, LTP-IE does not facilitate spiking since the membrane voltage remains subthreshold; for overly strong $w^{PC,G}$, spontaneous spike rates are already high, and increased spiking leads to more frequent voltage resetting [*Figure 1G*, *Figure 1—figure supplement 1*] so total spike rate does not substantially increase.) Thus, given moderate $w^{PC,G}$, when a cell undergoes LTP-IE its chance of spiking in response to future inputs (i.e. its excitability) increases, although variability remains. Since LTP-IE-triggering spike rates are only around 10–20 Hz, this suggests LTP-IE may play an active role in modulating firing properties of recently active cells in physiological conditions.

## Spike sequences propagate along LTP-IE-defined paths through a network

We next asked how LTP-IE as described in *Figure 1* would shape activity when introduced into a recurrently connected model network of 3000 excitatory spiking pyramidal cells (PCs). Intuitively, since LTP-IE increases spiking probability within recently active neurons, we expect recently active neurons to be preferentially recruited during sequence propagation through the network. Inspired by hippocampal place cells (*O'Keefe, 1979*; *Moser et al., 2008*), we allowed excitatory PCs in our network to be tuned to specific positions ('place fields') within the environment (*Figure 2A–C*). We furthermore assumed stronger connectivity between PCs with nearby place fields (*Figure 2D* and Materials and methods), such that propagation would be more likely to reflect movement along continuous 'virtual' trajectories through the environment, and in particular trajectories containing LTP-IE-tagged cells. Finally, the excitatory PC network was connected to a population of 300 inhibitory (INH) cells, which served to control the number of simultaneously active PCs. PC→INH and INH→PC connection probability was 0.5.

To investigate network activity shaped by recent sensorimotor sequences, we considered the following scenario, in analogy with typical experimental setups used to measure neuronal replay in hippocampus (*Foster and Wilson, 2006*; *Davidson et al., 2009*; *Carr et al., 2011*): a rodent has just run along a short trajectory through its environment and now rests for several seconds in an awake, quiescent state. How does the neural activity evoked during the trajectory shape the spontaneous activity in the awake, quiescent state? Since our goal was to understand sequential reactivation during this awake, quiescent period, we only used the trajectory geometry to predict the LTP-IE levels one would expect following the trajectory's termination (i.e. the termination of S inputs). To do so, we computed each neuron's maximum expected spike rate during the trajectory as a function of the

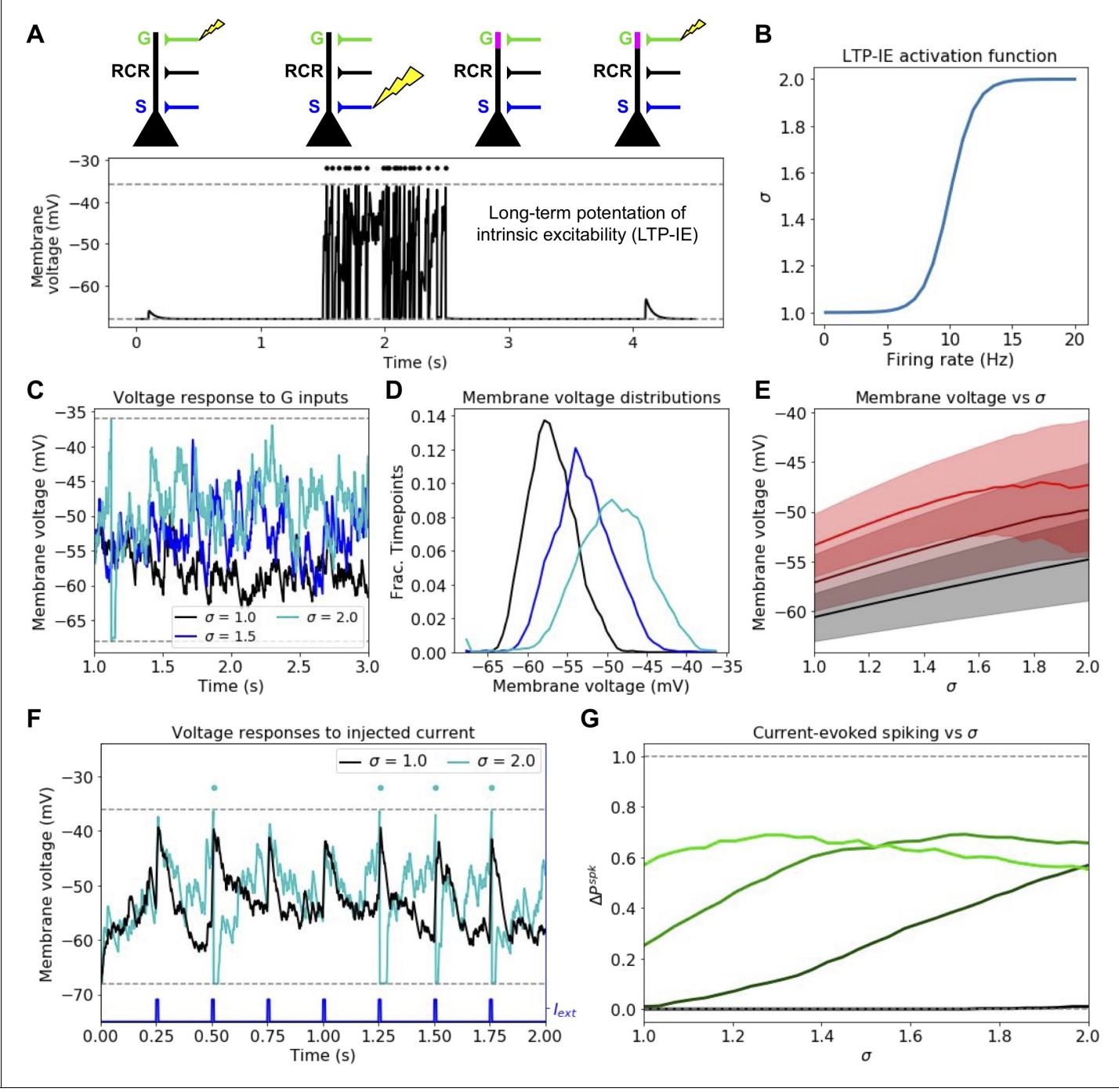

**Figure 1.** Mechanism and consequences of LTP-IE. (**A**) Demonstration using leaky integrate-and-fire (LIF) neuron model of fast activity-driven LTP-IE, which doubles membrane voltage responses to gate inputs as described in *Hyun et al. (2013)*; *Hyun et al. (2015)*. A spike in an upstream gate neuron (**G**) first elicits a small EPSP in the pyramidal cell (PC); a 1 s spike train (dots) at approximately 20 Hz is evoked by strong stimulation of sensory inputs S; when G spikes again, the EPSP has doubled in size. 'RCR' refers to recurrent inputs from other PCs (not used in this figure). Dashed lines show leak and spike-threshold voltages. (**B**) Shifted logistic function for LTP-IE strength (effective weight scaling factor) σ vs. PC firing rate over 1 s. (**C**) Example membrane voltages of PCs with different σ receiving stochastic but statistically identical gating input spikes. (**D**) Distribution of PC membrane voltage for σ values shown in C. (**E**) Mean (thick) and standard deviation (shading) of $V_m$ as a function of σ for three gate firing rates. Black: $r_G$ = 75 Hz; dark red: $r_G$ = 125 Hz; red: $r_G$ = 175 Hz. (**F**) Example differential sensitivities of PC spike responses to injected current input (blue) for two different σ. Dashed lines show leak and spike threshold potential; dots indicate spikes (which only occur for the σ = 2 case [cyan]). (**G**) Difference between current-evoked spike probability and spontaneous spike probability as a function of σ for four initial gate input weights. Color code, in order of increasing lightness: $w^{PG,G}$ = 0.4,. 8, 1.2, 1.6 (see Materials and methods for units).

*Figure 1 continued on next page*

*Figure 1 continued*

DOI: https://doi.org/10.7554/eLife.44324.002

The following figure supplement is available for figure 1:

**Figure supplement 1.** Effects of current pulses on spiking under LTP-IE.

DOI: https://doi.org/10.7554/eLife.44324.003

distance from its place field to the nearest point on the trajectory (*Figure 2C*). We then passed the result through a soft-thresholding LTP-IE activation function (*Figure 1B*) to compute the final expected LTP-IE level σ (*Figure 2C*) for each PC. This allowed us to model the expected LTP-IE profile over the network of neurons as a function of the recent trajectory (*Figure 2E*). As per our design, the LTP-IE profile stores which locations were covered by the original trajectory but bears no explicit information about its speed or direction.

Can replay-like sequences that recapitulate the original trajectory structure emerge from the LTP-IE profile stored in the network? When we let our network run in the presence of random G inputs independent to each PC, replay-like events lasting on the order of 100–200 ms spontaneously arose in the network (*Figure 2F–I*). Spontaneous replay events propagated in both directions (*Figure 2F*, left and middle), recapitulating the forward and reverse replay observed experimentally (*Foster and Wilson, 2006*; *Davidson et al., 2009*; *Carr et al., 2011*). Due to the PC refractory period of ~8 ms in our model (see Materials and methods), activity propagated in one direction without reversing. Partial replay also occurred, in which replay began in the middle of the trajectory and propagated in one direction to the end (*Figure 2F*, right). When an alternative trajectory was used to induce the LTP-IE profile, triggered replay recapitulated that trajectory instead (*Figure 2—figure supplement 1A–E*), indicating that replay of a specific trajectory was not due *a priori* to the recurrent connectivity; indeed, this should be avoided by the spatial uniformity of the recurrence. The virtual trajectory encoded by the replay event could be decoded by computing the median place field of the PCs spiking across short time windows during the event (*Figure 2—figure supplement 1F*). We additionally note that the replay sequence was able to turn corners, suggesting robustness to nonlinear trajectories. Finally, plotting population firing rates (*Figure 2I*) reveals a potential connection between the replay events in our model and 'sharp-wave ripples', rapid oscillatory field potentials experimentally observed to co-occur with hippocampal sequence replay (*Carr et al., 2011*). Thus, despite sizable variability in membrane voltages arising from the random gating inputs G (*Figure 1C,D,F*), LTP-IE was able to induce spontaneously arising activity sequences into the network that recapitulated recent sensorimotor experiences.

## Dependence of LTP-IE-based sequence propagation on network parameters

What conditions must hold for LTP-IE to induce successful sequences in the network? To address this we explored how the frequency of spontaneous replay events changed as we varied different network parameters. To ensure that we only analyzed replay events reflecting recent sensorimotor experiences, we only included events in which average PC activity was sufficiently elevated and in which PC spikes were confined primarily to LTP-IE-tagged PCs.

We observed that spontaneous replay events among LTP-IE-tagged cells arose across a sizeable range of network parameters (*Figure 3A–C*). We first found that replay events occurred with a frequency greater than 1 Hz across a large range of excitatory and inhibitory feedback strengths, $w^{PC,PC}$ and $w^{PC,INH}$, respectively (*Figure 3A*); when $w^{PC,PC}$ was too large without sufficient inhibitory compensation, however, the network entered a 'blowup' regime in which activity spread across the whole network instead of being confined to the LTP-IE-tagged PCs (*Figure 3A*, black). A further role for inhibition is demonstrated in *Figure 3D–E*. In particular, while replay confined to LTP-IE-tagged PCs can occur in the absence of inhibitory feedback (*Figure 3D*-right), in this case one observes an increase in spontaneous 'bidirectional' events, in which replay activity begins in the middle of the trajectory and propagates outward in both directions (compare *Figure 3D*-left to *Figure 2F*-right).

In addition to being robust to changes in $w^{PC,PC}$, spontaneous replay events also arose across a substantial range of recurrent connectivity length scales $\lambda^{PC,PC}$ (*Figure 3B*). Notably, $\lambda^{PC,PC}$ and $w^{PC,PC}$ both shaped the 'virtual relay speed' of the replayed trajectory, with increases in either parameter

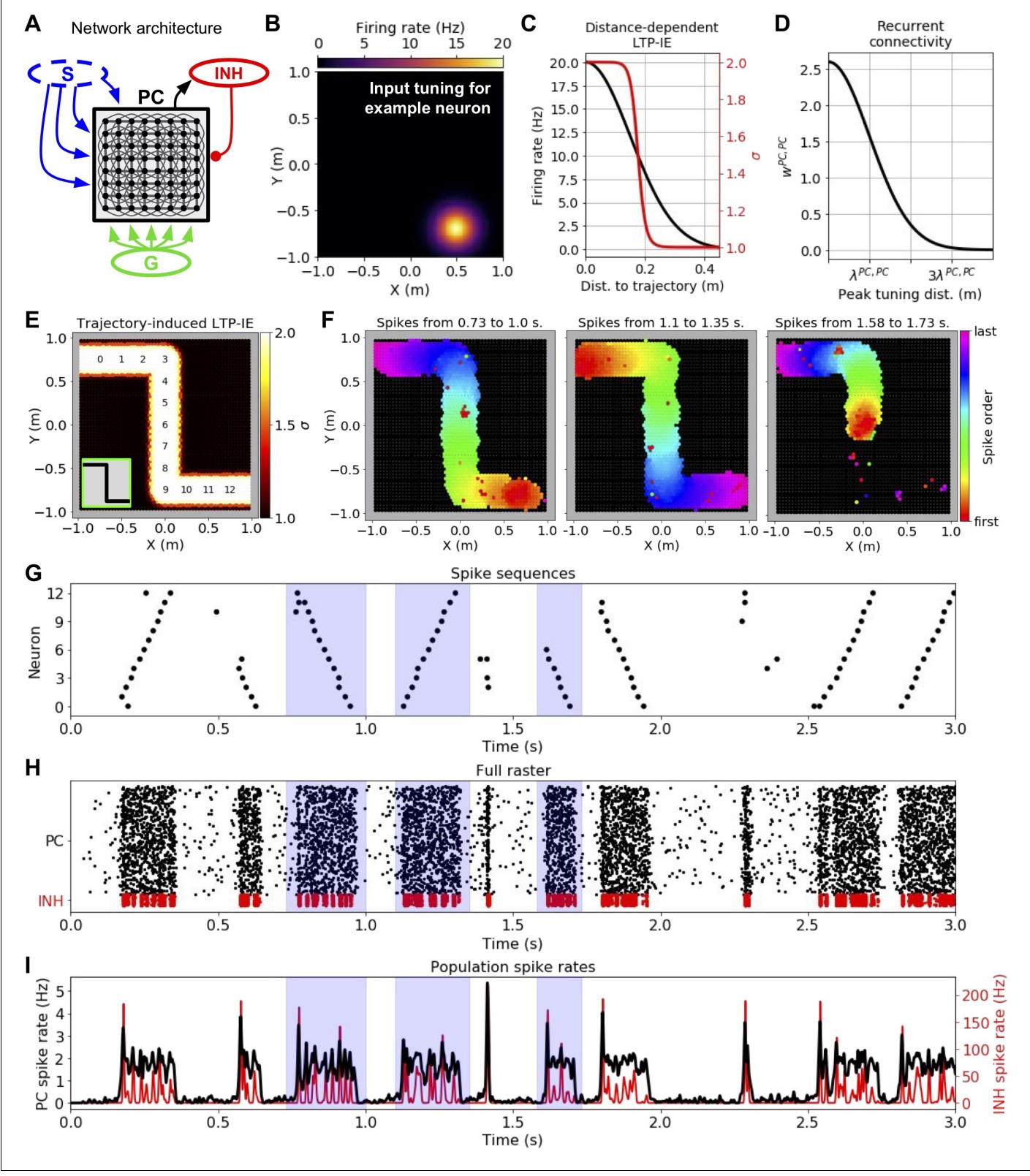

**Figure 2.** Demonstration of history-dependent sequence replay via LTP-IE. (**A**) Network architecture used in simulations. The dashed oval around input S indicates that we did not explicitly simulate its interaction with the pyramidal cell (PC) network, but rather only the LTP-IE profile one would expect following the termination of S input. (**B**) Squared exponential position tuning for an example neuron (20 Hz max. firing rate, 0.15 m length scale). (**C**) Resultant LTP-IE as a function of distance between a cell's maximum tuning and the closest point on the trajectory through the environment (red),

*Figure 2 continued on next page*

*Figure 2 continued*

computed as a sigmoidal function (***Figure 1B***) of position-dependent firing rate (black). (**D**) Recurrent excitatory weights between PCs as squared exponential function of distance between the two cells' peak tuning positions. (**E**) LTP-IE profile induced in PCs by an example Z-shaped trajectory (inset) in a network of 3000 PC and 300 inhibitory (INH) cells. PCs are positioned according to peak tuning and colored by the LTP-IE level ($\sigma$) expected to result from the trajectory. The numbers indicate the position tunings of the PC identifiers in G. (Note: individual cells cannot be seen here due to their high density). (**F**) Cells activated during the different spontaneous replay events shaded in blue in G-I, colored by the order of the first spikes each cell emitted during the event. Black cells did not activate during replay. Left and middle: replay in two different directions; right: partial replay. G. Partial raster showing spike times for cells with position tunings marked in E. H. Full raster plot for PC and INH population over a 3 s trial. I. Cell-averaged spike rates for PC and INH populations throughout trial.

DOI: https://doi.org/10.7554/eLife.44324.004

The following figure supplement is available for figure 2:

**Figure supplement 1.** LTP-IE-based replay for an alternative trajectory/sequence to that shown in ***Figure 2***.

DOI: https://doi.org/10.7554/eLife.44324.005

generally yielding faster replay (***Figure 3F–G***). This result corroborates the hypothesis that the speed of replay events emerges from internal network structure, as opposed to the temporal structure of the behavioral trajectory (***Davidson et al., 2009***). Spontaneous replay events also occurred reliably across a range of maximal LTP-IE levels ($\sigma_{max}$) and random gating input rates $r^G$ (***Figure 3C***), suggesting that neither of these variables would have to be precisely tuned in vivo for successful sequence replay.

We next investigated two other aspects of our sequence replay model: sharp-wave ripples (SWRs) and sequence propagation with branched LTP-IE profiles. Inspired by the resemblance of the population firing rates accompanying LTP-IE-induced replay in our network model to experimentally observed sharp-wave ripples (***Figure 2I***) (***Csicsvari et al., 1999***; ***Carr et al., 2011***), we computed the power spectral density (PSD) of the population PC firing rates accompanying replay events for networks constructed with two different parameter sets that differed in their level of inhibitory feedback (***Figure 3H***). The resulting PSDs showed salient peaks at tens of Hz, suggesting replay in our network is accompanied by oscillatory SWR-like events. While empirical SWRs in CA3 occur near higher frequencies of 100–130 Hz (***Csicsvari et al., 1999***), the two networks we investigated also exhibited different PSD peaks, with the more strongly inhibited PC population displaying a higher peak frequency. This suggests the SWR oscillation frequencies in our model may be determined by the balance of excitatory and inhibitory feedback strengths and that parameter regimes may exist where SWRs alongside replay occur at biological frequencies.

Since realistic locomotor trajectories often intersect with one another, we also investigated how our model behaved when the trajectory reflected in the LTP-IE profile contained multiple pathways branching out from a central intersection (***Figure 3I***). In this scenario we observed that spontaneous replay events in our model could travel through the intersection, selecting a single path along which to continue propagating (***Figure 3J***, left and middle). When inhibition was removed from the model, LTP-IE-induced spike sequences propagated along all branches after the intersection (***Figure 3J***, right), suggesting inhibition likely imposes a winner-take-all-like interaction among competing sequences so that only one sequence continues propagating.

These results show that experience-dependent LTP-IE at biological levels can rapidly and selectively modulate sequence propagation through a network of realistic spiking neurons across a range of parameter regimes.

## LTP-IE-based sequences can encode temporary stimulus-response mappings

A popular hypothesis in cognitive neuroscience is the multiplexing and interaction of mnemonic and spatial representations, contributing, for example, to the formation of 'memory maps'. Indeed, hippocampus' experimentally observed roles in both memory and spatial navigation lends compelling evidence to this notion (***Schiller et al., 2015***). The potential neural mechanisms underlying this phenomena, however, remain poorly understood. To demonstrate how LTP-IE could shape temporary memory storage and computation more generally, beyond simply replaying recent locomotor trajectories, we considered the simple task of requiring a network to represent one of two possible stimulus-response mappings, and then asked how LTP-IE could induce these mappings in the network

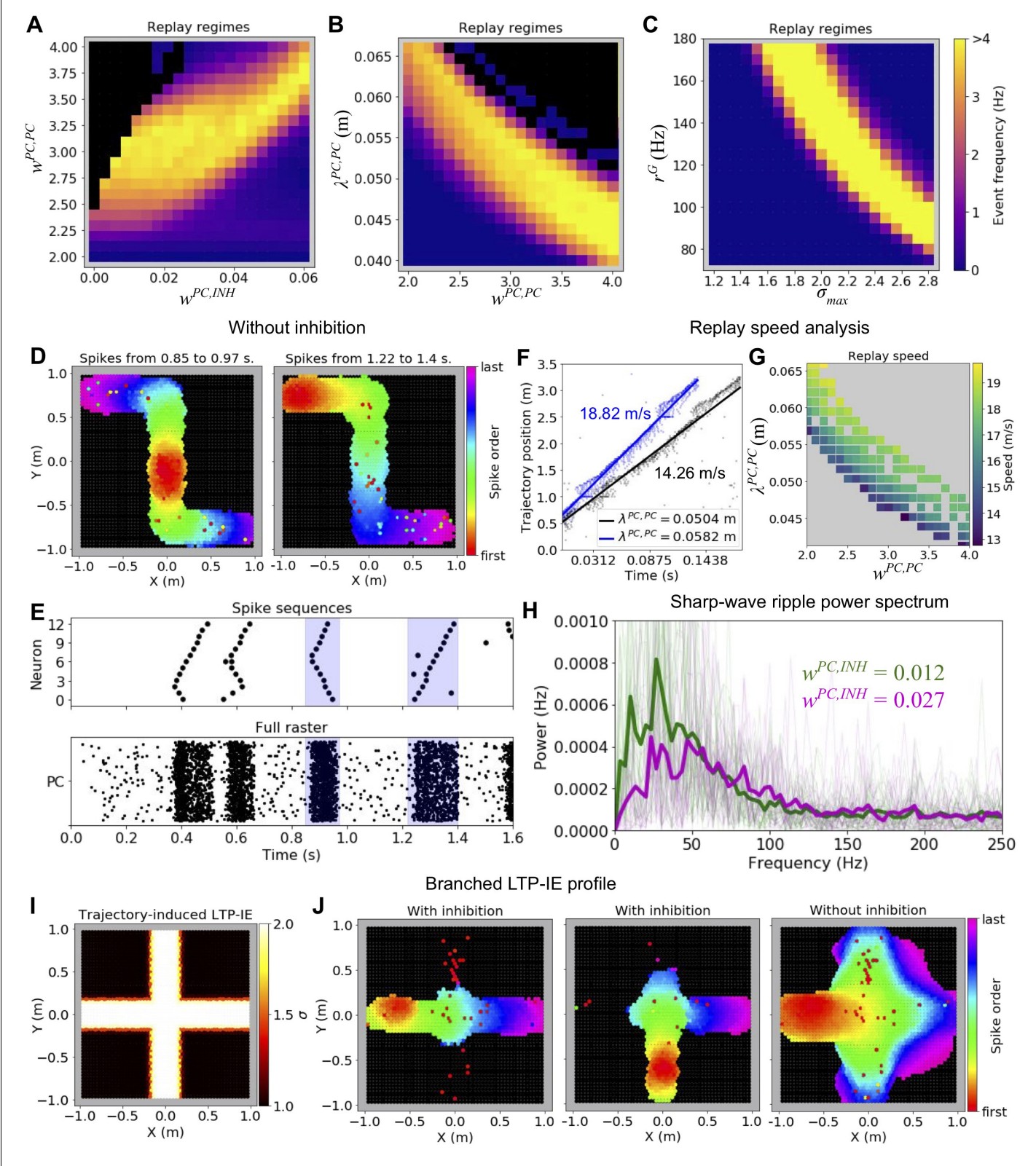

**Figure 3.** Parameter dependence of LTP-IE-based sequences. (**A**) Spontaneous replay event frequency as a function of recurrent excitation ($w^{PC,PC}$) and inhibitory feedback ($w^{PC,INH}$) strengths. Black regions indicate regimes of 'blowup' replay events, which were not confined to LTP-IE profile. Weights are given in units of the conductance change in response to a presynaptic spike, relative to the leak conductance (inhibitory feedback weights are subsantially smaller because of the high connection probability between PCs and inhibitory cells). (**B**) As in A, but as a function of excitatory connectivity

*Figure 3 continued on next page*

*Figure 3 continued*

length scale ($\lambda^{PC,PC}$) and $w^{PC,PC}$. (C) As in A, B, but as function of gate input frequency ($r^G$) and the maximum LTP-IE level ($\sigma_{max}$) a cell could attain. (D) Sequential firing of active cells (black cells did not spike) during replay events in a network without inhibition. (E) Partial (top) and full (bottom) raster plots showing replay event structure in network without inhibition. Cells in top raster are ordered by tuning positions indicated in 2E. (F) Virtual replay speed calculations for spontaneous replay events in networks with two different connectivity length scales $\lambda^{PC,PC}$. Point clouds show linear position of place fields of spiking PCs vs their spike times during replay. Thick lines are best-fit regression lines. (G) Estimated replay speed as a function of $\lambda^{PC,PC}$ and $w^{PC,PC}$. (H) Power spectral density of PC population activity during replay events for two different levels of inhibitory feedback $w^{PC,INH}$. Thin lines correspond to individual replay events, and thick lines to event averages. (I) Example branched LTP-IE profile. (J) Spike order during spontaneous replay events given branched LTP-IE profile in I. Left and middle: example events in network with inhibition; right: example event in network without inhibition.

DOI: https://doi.org/10.7554/eLife.44324.006

(*Figure 4A*). We assume such an induction could be evoked by appropriately transformed sensori-motor or cognitive inputs from upstream areas, but we did not model this explicitly, as our goal was to demonstrate the final storage and recall of the mapping. Note that in the following analysis we have used a larger $r^G$ (200 Hz) and decreased $w^{PC,G}$ (0.5) and $\sigma_{max}$ (1.84), to reduce spontaneous activity while still allowing propagation of sequences triggered by a brief current injection.

LTP-IE was able to induce multiple stimulus-response mappings in the network, each in multiple ways. To demonstrate this we first assumed that specific regions in the network corresponded to different stimuli or responses. In other words, we imagined that stimulus regions S1 and S2 might receive inputs containing sensory information from the environment, whereas the response regions M1 and M2 might send outputs controlling different motor commands (*Figure 4B*). To record activity in S1, S2, M1, or M2 we included a unique readout unit for each region representing the region's average activity (*Figure 4B*). We then introduced non-intersecting LTP-IE paths into the network connecting each stimulus region to its associated motor output region, and recorded the ability of an input current trigger into each stimulus region to subsequently activate the correct motor output (*Figure 4C–F*). We first verified our idea with the mapping (S1→M1, S2→M2) in which each stimulus was connected via a one-dimensional LTP-IE profile to its appropriate motor output (*Figure 4C*, left). Indeed, triggering each stimulus region with a depolarizing current input led to subsequent activation of its corresponding motor output after a short delay (*Figure 4C*, middle and right). We next explored the mapping (S1→M2, S2→M1), which in our setup required at least one LTP-IE path to take a roundabout course through the network (*Figure 4D*, left). As before, triggering each stimulus region led to activation of its corresponding motor output, with the longer LTP-IE path reflected in a longer stimulus-response delay (*Figure 4D*, middle and right). Notably, the same stimulus-response mapping could be implemented via LTP-IE in several different ways (*Figure 4D–F*), with the different LTP-IE paths reflected in the different stimulus-response delays. This contrasts with alternative models of temporarily binding together distinct components of a neural network, which typically suppose a unique structure of the mapping/binding representation (*Raffone and Wolters, 2001*; *Botvinick and Watanabe, 2007*; *Swan and Wyble, 2014*). Thus, LTP-IE combined with recurrent connectivity might serve as a biophysically plausible and highly flexible substrate for inducing temporary stimulus-response mappings in a recurrent spiking network. This may be a potential key property enabling rapid and flexible induction of temporary information-processing patterns in brain networks underlying cognitive flexibility.

## Encoding complex sequences and non-spatial mappings with LTP-IE

What are the algorithmic limitations of LTP-IE-based sequence induction? For instance, while we have discussed the storage and reactivation of sequences corresponding to simple (non-intersecting) trajectories through space, it is natural to ask whether more arbitrary sequences and mappings can be stored by LTP-IE. To this end we developed a reduced network model capturing the core property of LTP-IE-based reshaping of network dynamics. This allowed us to separate the generic algorithmic properties of LTP-IE- and excitability-based computation from those tied to its specific biophysical implementation.

Our reduced network obeys the following rules. First, it operates in discrete time, with upstream inputs to a neuron persisting for one timestep, and in which a neuron spikes if those inputs surpass a spike threshold. When a neuron spikes, this spike is converted to inputs to downstream neurons, weighted by the connectivity strengths to those neurons (only excitatory connections were used in

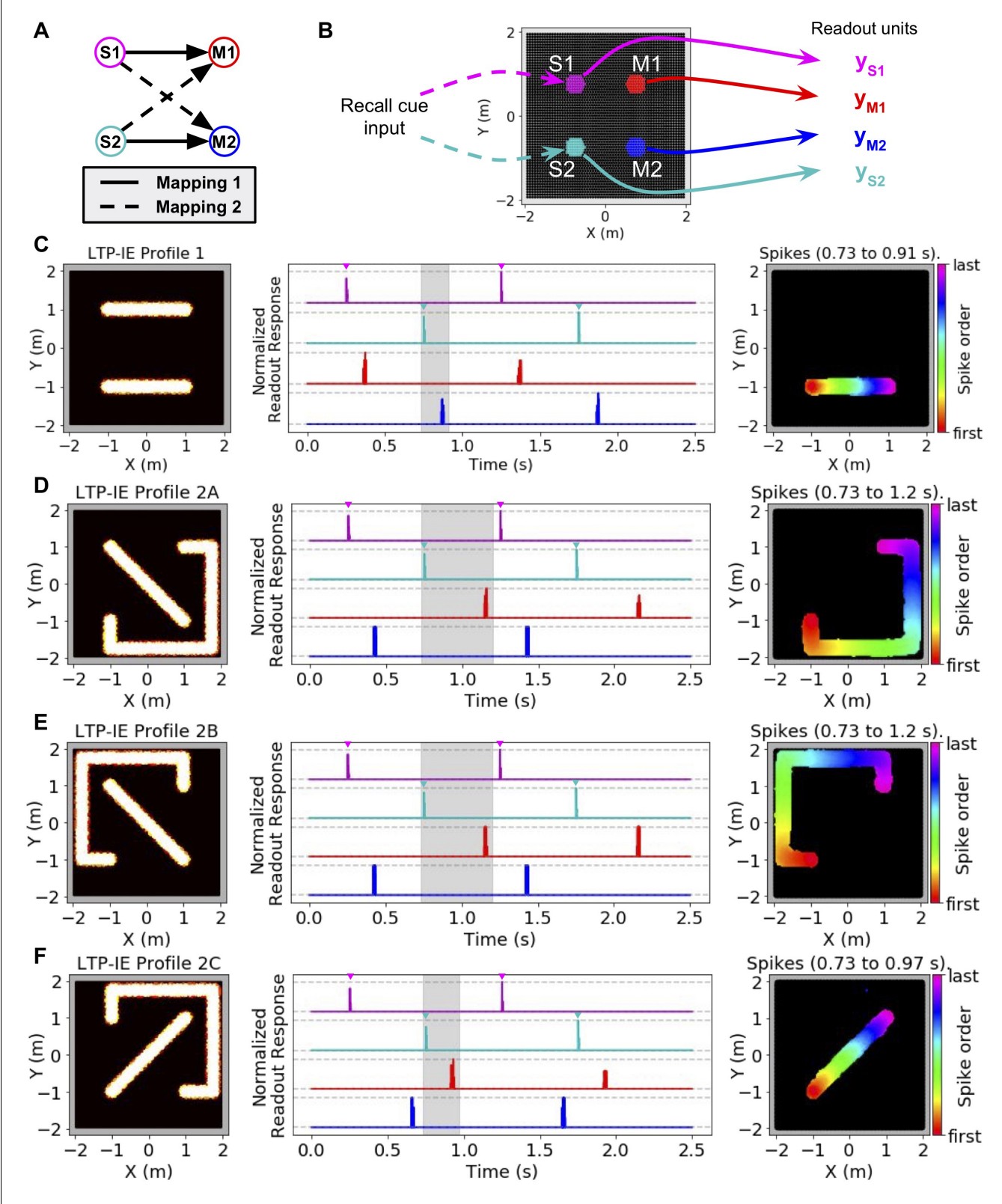

**Figure 4.** Using LTP-IE-based sequential activity propagation to maintain and decode pairwise associations. Here we used a network with increased $r^G$ (200 Hz) and decreased $w^{PC,G}$ (0.5) and $\sigma^{max}$ (1.84), which decreased spontaneous activity without affecting propagation of sequences triggered by direct current injection (see Materials and methods). (**A**) Example of two different mappings between a pair of stimuli (S1, S2) and a pair of responses (M1, M2). In Mapping 1 (solid), activating S1 should activate M1, and activating S2 should activate M2. In Mapping 2 (dashed) activating S1 should

*Figure 4 continued on next page*

*Figure 4 continued*

activate M2, and activating S2 should activate M1. (**B**) Simulation protocol. Either S1 or S2 is activated with a recall cue (injected current input into colored cells). Readout units average all activity from either S1, S2, M1, or M2 units, indicated by colors. (**C**) Left: Example LTP-IE-based encoding of Mapping one with neurons laid out in 2-D space. Middle: Time-dependent readout responses (normalized to maximum readout response over the 2.5 s simulation), with colors corresponding to readout units depicted in B. S1 and S2 were each alternately activated 3 times by appropriate recall cue (direct current injection into the relevant neurons is indicated by the inverted colored triangles), and all readout responses were plotted. In this example, activating S1 (magenta) causes M1 (red) to activate, and activating S2 (cyan) causes M2 (blue) to activate, due to spike propagation along paths defined by LTP-IE profile in C-Left. Right: Order of first spikes of all neurons that spiked during shaded time period in C-Middle. (**D**) Example LTP-IE-based encoding of Mapping two with neurons laid out in same 2-D coordinate space as in C, along with readout responses and spike order during shaded epoch. (**E**) Same as C-D but for an LTP-IE profile encoding Mapping two that is distinct from that in D. (**F**) Same as D and E but for a third LTP-IE profile encoding Mapping 2.

DOI: https://doi.org/10.7554/eLife.44324.007

our reduced network model), and the spiking neuron enters a refractory period that prevents it from spiking again for the next several timesteps. To model gating inputs G and LTP-IE in our reduced model, we provided all neurons with a constant background input representing G, with LTP-IE-tagged neurons receiving up to twice this background input, thus situating them closer to spike threshold. To mimic inhibitory effects we imposed a maximum number of simultaneously spiking cells, chosen to be those receiving the largest inputs.

Using our reduced model we first demonstrate how LTP-IE could support the storage of *complex* trajectories, in which one of the replayed locations is repeated, that is trajectories that intersect themselves (*Figure 5A*, inset). Importantly, here we desire replay without random branching at the intersection point, as one would expect given *Figure 3J*. To overcome this we introduced an additional tuning dimension to our network. Beyond an (x, y) position, we allowed each cell to be Gaussian-tuned to head-direction θ as well, in line with experimentally observed head-direction cells in the hippocampus (*Leutgeb et al., 2000*), and we only tagged PCs with LTP-IE if they both (1) lay along the trajectory and (2) had a preferred θ aligned to the animal's presumed travel direction during the original locomotor trajectory. Our assumption is that all three tuning conditions would have to be met during the original trajectory in order for these cells to activate at the physiological levels required for LTP-IE. Accordingly, we modified the recurrent connectivity by only preferentially connecting cells if they both encoded nearby regions of space and had a similar preferred θ.

The LTP-IE profile of an example self-intersecting trajectory is shown in *Figure 5A*. This trajectory begins eastbound at (−1,. 25) m, loops around through the lower right quadrant of the environment, crosses itself at (−0.25,. 25) m, and ends northbound at (−0.25, 1) m. Importantly, at the intersection point, only east- and north-tuned cells exhibit strong LTP-IE, and since these cells are not strongly connected, activity propagating through the east-tuned cells should not spread to north-tuned cells, and vice versa. Spike sequence propagation through the network triggered by a brief current injection into cells at the beginning or end of the trajectory is shown in *Figure 5B,C*, respectively. Activity recapitulating the original trajectory propagated in either the forward or reverse direction, depending on the trigger location, and activity passed through the intersection point in the correct order, without random branching. Note that one would expect random branching probability to increase as the intersection became less orthogonal. This demonstrates how LTP-IE combined with multidimensional tuning and corresponding recurrent connectivity can support trajectory sequences containing repeated locations, indicating that LTP-IE-based replay need not be confined to simple sequences.

Finally, we use our reduced model to show how LTP-IE can encode transient stimulus-response mappings without requiring a spatially organized network. While hippocampus has been extensively implicated in spatial computations (*Moser et al., 2008*; *Hartley et al., 2014*), exploring how LTP-IE could shape computation without such structure may shed light on how excitability-based computations could arise in other brain networks more generally. To this end we again considered the problem of rapidly inducing one of two sensorimotor mappings (S1→M1, S2→M2) vs (S1→M2, S2→M1) in a network through LTP-IE. Here we represented S1, S2, M1, and M2 as four separate neural ensembles, with the M ensembles containing strong intra-ensemble recurrent excitation but with no recurrence between ensembles (*Figure 5D,E*). Instead, each sensory ensemble sent random projections to a 'switchboard' ensemble of neurons, and the switchboard ensemble sent random

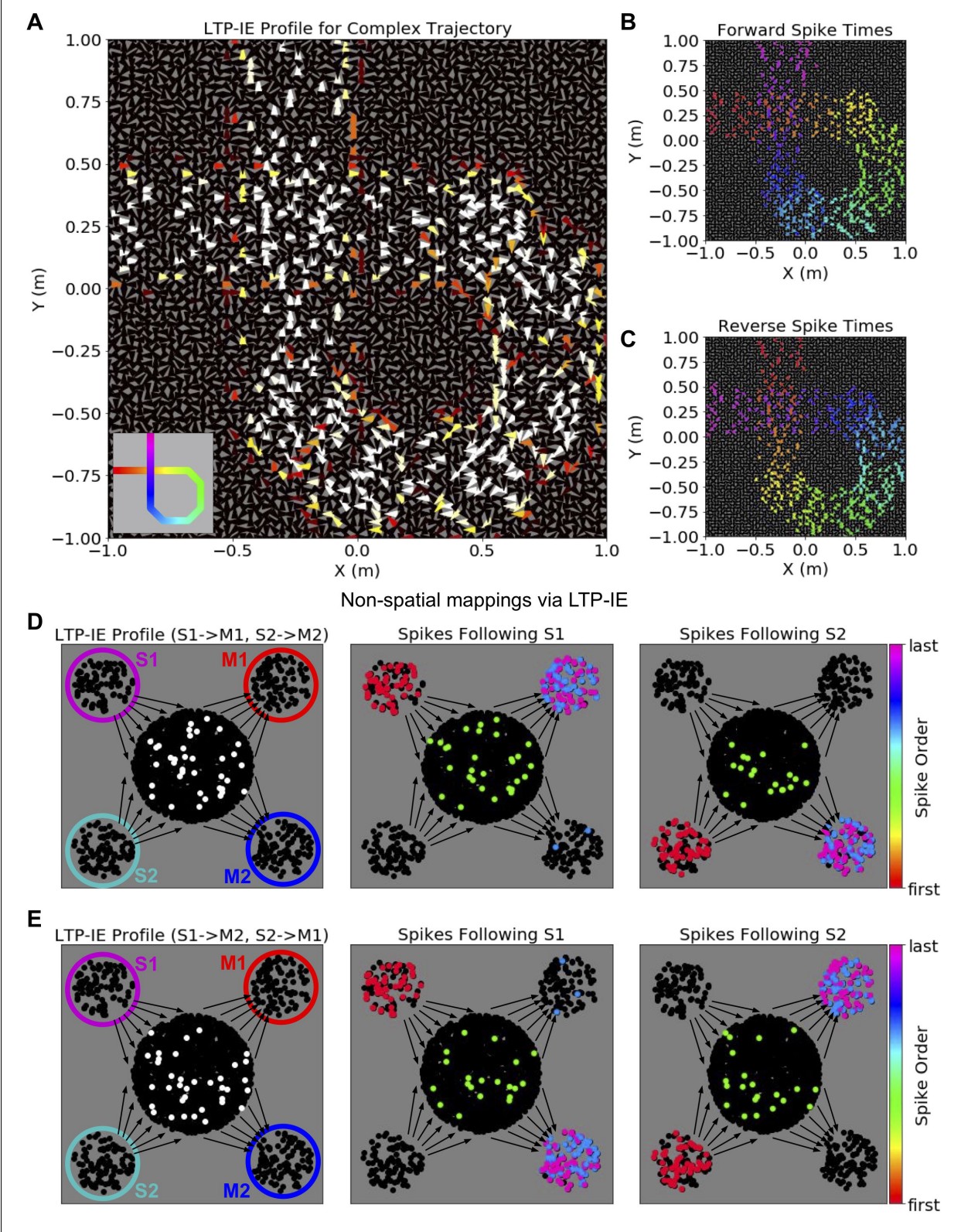

**Figure 5.** Complex sequences and non-spatial mappings via LTP-IE in our reduced network model. (**A**) LTP-IE profile for self-intersecting trajectory in a network with cells tuned to both position and head direction. Each triangle corresponds to a single cell, with location indicating position tuning and triangle orientation indicating head-direction tuning. This LTP-IE profile corresponds to a 'gamma'-shaped trajectory (inset) beginning at (−1,. 25) m and ending at (−0.25, 1) m, and which intersects itself at (−0.25,. 25) m. (**B**) 'Forward' cell activation sequence given the network and LTP-IE profile in A,

*Figure 5 continued on next page*

*Figure 5 continued*

triggered by current injection into cells with tuning near (−1,. 25) m. (**C**) 'Reverse' cell activation sequence given the network and LTP-IE profile in A, triggered by current injection into cells with tuning near (−0.25, 1) m. Colors in B, C indicate spike order, with same color code as in 2F. (**D**) Left: LTP-IE profile for one mapping between two stimuli (S1, S2) and two responses (M1, M2) in a network without spatial organization. (The only neurons tagged by LTP-IE were the ones shown in white, which had an LTP-IE level of σ = 2.) Each stimulus (magenta, cyan) or response (red, blue) is represented by an ensemble of 100 neurons, with the 'switchboard ensemble' containing 2000 neurons (recurrence with M1, M2 exists but is not depicted here); middle: spike order following current injection into the S1 neurons, leading to subsequent activation of M1; right: spike order following current injection into the S2 neurons, leading to subsequent activation of M2. (**E**) Left: As in D-Left, but with an LTP-IE profile encoding the opposite mapping from D; middle: spike order following current injection into the S1 neurons, leading to subsequent activation of M2; right: spike order following current injection into the S2 neurons, leading to subsequent activation of M1.

DOI: https://doi.org/10.7554/eLife.44324.008

connections to the motor ensembles (*Figure 5D,E*). For the purposes of this simulation we did not include recurrence within the S or switchboard ensembles.

Given this network architecture, we encoded each sensorimotor mapping by applying LTP-IE to different sets of switchboard neurons. For (S1→M1, S2→M2) we applied LTP-IE to all switchboard neurons that either (1) received inputs from S1 *and* sent outputs to M1 or (2) received inputs from S2 *and* sent outputs to M2 (*Figure 5D*). Intuitively, activating S1 should activate the first set of LTP-IE-tagged neurons, causing M1 to activate, whereas activating S2 should activate the second set of LTP-IE-tagged neurons, causing M2 to activate. This is shown in *Figure 5D* (**middle and right**), where activating each stimulus indeed activates a subset of LTP-IE-tagged switchboard neurons, causing the appropriate motor ensemble to subsequently activate. Notably, the alternate mapping (S1→M2, S2→M1), in which LTP-IE is applied to switchboard neurons that either (1) receive inputs from S1 *and* send outputs to M2 or (2) receive inputs from S2 *and* send outputs to M1 (*Figure 5E*), yields an LTP-IE profile distinct from the one encoding (S1→M1, S2→M2); instead, this LTP-IE profile allows signal propagation from S1 to M2 and from S2 to M1. We note that the fundamental idea of opening selective communication channels between pairs of ensembles by changing the spiking properties of an intermediary set of switchboard neurons has been explored in substantial detail in the quite similar 'binding pool' model of *Swan and Wyble (2014)*. Here we show how the idea also fits naturally into the theory of excitability-based computation. We do not explore how the specific LTP-IE profiles above could be induced in the first place, assuming instead that they could arise via upstream transformations of sensorimotor information. Thus, even in a randomly connected network with no spatial structure, LTP-IE can yield selective stimulus-response mappings, demonstrating the potential of excitability changes as a substrate for cognitive flexibility more generally.

## Discussion

Sequential spiking activity is a key feature of recurrent neural network dynamics, potentially reflecting information flow and computations within the network. One noteworthy empirical example of internally generated sequences is the replay of spike sequences representing an animal's recent sensorimotor sequence, such as a locomotor trajectory or sequence of viewed images. Sequence replay has been observed in vivo in both hippocampus and cortex during awake quiescent periods in both rodents and primates (*Foster and Wilson, 2006*; *Davidson et al., 2009*; *Karlsson and Frank, 2009*; *Gupta et al., 2010*; *Carr et al., 2011*; *Eagleman and Dragoi, 2012*) and is thought to be involved in memory consolidation (*Carr et al., 2011*; *Jadhav et al., 2012*; *Ólafsdóttir et al., 2017*; *Zielinski et al., 2017*) and navigational planning (*Foster and Knierim, 2012*; *Pfeiffer and Foster, 2013*). Since replay appears to be strongly modulated by recent experience, it serves as a compelling entrypoint for uncovering the biophysical mechanisms that support rapid and flexible modulation of sequential dynamics and information processing in neural networks.

While substantial work has shown how gradual modifications to a network's recurrent connections can induce sequences into its dynamics repertoire (*Sussillo and Abbott, 2009*; *Klampfl and Maass, 2013*; *Laje and Buonomano, 2013*; *Rajan et al., 2016*), less is known about mechanisms that could act on faster timescales. Here we have demonstrated the sufficiency of an empirical, strong, and fast-acting heterosynaptic plasticity mechanism known as long-term potentiation of intrinsic excitability (LTP-IE) (*Hyun et al., 2013*; *Hyun et al., 2015*), applied to a spiking network, to coerce the

network into generating selective sequences. Briefly, LTP-IE biases recently active cells towards recruitment during replay, with their reactivation order determined by pre-existing but fixed recurrent connectivity. In particular, in our spatially organized spiking network LTP-IE acted to select pathways through the network that guided activity propagation during replay. Similar to 'synfire chains', feedforward neural networks supporting stable activity propagation (*Abeles, 1982*; *Abeles et al., 1994*; *Herrmann et al., 1995*; *Schrader et al., 2008*), the potentiated pathways in our network supported stable and self-sustaining spike sequences, which despite symmetric connectivity propagated without reversing direction due to the moderate refractory period we included (8 ms). One can therefore conceptualize our spatial network as akin to a reservoir of trajectories through space that can be selected for replay in an experience-dependent manner through LTP-IE.

Our spiking network model exhibited several interesting features. First, despite substantial variability in the membrane voltages of LTP-IE-tagged cells and overlap with non-LTP-IE-tagged cells (*Figure 1D*), replay events exhibited stable propagation along LTP-IE-defined network paths, due to averaging effects of populations of LTP-IE-tagged cells. Second, sequences replayed at timescales determined by the network rather than the original trajectory (*Figure 3F*), and did so in both forward and reverse (*Figure 2F,G*), capturing these two key empirical features of hippocampal replay (*Foster and Wilson, 2006*; *Davidson et al., 2009*; *Karlsson and Frank, 2009*; *Gupta et al., 2010*; *Carr et al., 2011*). Additionally, inhibition played an important role in our model, promoting replay in one direction only (*Figure 2F*) and inducing competition among multiple sequences in branched trajectories (*Figure 3D,I,J*), even though inhibition bore no spatial organization itself. Inhibition also appeared to shape the power spectra of the sharp-wave-ripple (SWR) events in our model arising in the PC population average alongside replayed sequences; moreover, these events were consistent with the observation that hippocampal SWRs likely arise spontaneously in CA3 and propagate to CA1 (*Buzsáki, 1986*; *Csicsvari et al., 2000*; *Carr et al., 2011*; *Ramirez-Villegas et al., 2018*), since our model's excitatory recurrence was inspired by that found in CA3 but not CA1 (*Lisman, 1999*).

Finally, while our biophysical model used a 2-D spatial network and hippocampal LTP-IE mechanism, excitability changes might support more general modulation in dynamics and computation in networks with fixed recurrence. To explore this we used a reduced network and LTP-IE model attempting to capture the core algorithmic features of excitability-modulated dynamics independently from a specific biophysical implementation. For instance, LTP-IE might also occur purely intrinsically, without an explicit correlation with EPSP sizes (*Ohtsuki and Hansel, 2018*). Using the reduced model we showed how trajectories that intersect themselves in 2-D could be successfully replayed given an additional tuning dimension like head direction (*Figure 5A–C*), and further how LTP-IE could store and recall sensorimotor mappings in a randomly connected network with no spatial structure at all (*Figure 5D–E*), similar to the 'binding pool' model of *Swan and Wyble (2014)*. This latter example may be relevant to excitability-based computations in cortex, where LTP-IE-like phenomena have been observed (*Sourdet et al., 2003*; *Paz, 2009*), but whose internal network structures do not necessarily exhibit clear spatial organization. We note, however, that preferential connectivity among similarly tuned cells has been observed in mouse visual cortex (*Ko et al., 2011*), suggesting a mechanism similar to the one we have explored might support spontaneous sequence propagation among neurons encoding visual space. Such spontaneous sequences might in turn modulate neural responses to incoming visual stimuli, potentially supporting expectation-modulated sensory processing.

History-dependent excitability changes might also occur not just at the rapid timescales that have motivated our work but over slower timescales as well (*Zhang and Linden, 2003*; *Titley et al., 2017*). For example, it was recently found that small neuronal ensembles could be 'imprinted' into a mouse cortical network and subsequently reactivated in vivo via repeated optogenetic stimulation (*Carrillo-Reid et al., 2016*). While this result was suggested to provide evidence of Hebbian (homosynaptic) plasticity, our work suggests that the same result could arise via increasing neural excitabilities instead. Untangling these mechanisms and understanding how they interact presents an exciting avenue for further investigation. Overall, our work demonstrates how excitability changes alone might quickly and selectively reshape network dynamics, implicating their potential role in storing memories and encoding stimulus-response mappings, and more generally in organizing flexible computations over rapid timescales.

## Comparison to existing models

While we do not aim to provide a complete account of hippocampal replay, but rather to demonstrate how specific biophysical mechanisms can rapidly generate selective network sequences, we believe it is still worthwhile to compare our model to existing models for rapidly storing and recalling sequences in neural systems, both in terms of biological plausibility and computational robustness. One family of models for encoding and decoding sequences in hippocampus supposes that sequential information is stored in the timing of spikes relative to theta (4–8 Hz) and gamma (~40 Hz) oscillatory cycles in the hippocampus (*Lisman and Idiart, 1995*; *Lisman, 1999*). Briefly, gamma cycles are 'nested' within theta cycles, and a sequence of stimuli is stored by stimulus-specific neurons spiking at unique gamma cycles within the encompassing theta cycle. While these models provide a persuasive computational account of how oscillations might be used to store temporary information, it is not clear how the sensory input would be appropriately transformed so as to be 'entered' into the oscillation cycle at the correct time, how stable these mechanisms are in the face of noise, nor how multiple sequences might be stored simultaneously; in particular, sequence information could be lost if the oscillation were disrupted, and one would require an independent nested oscillation for a second sequence. In our LTP-IE-based spiking-network model, however, memories are directly entered into the network by physiological spiking patterns (*Figure 1*), are stored in effective synaptic weight changes likely driven by ion channel inactivation (as opposed to persistent spiking activity) and might therefore be more robust to noise, and multiple sequences can theoretically be stored in the same network in a manner similar to how we have stored multiple stimulus-response mappings in *Figure 4*.

The second main family of replay models either explicitly (*Molter et al., 2007*; *Veliz-Cuba et al., 2015*; *Haga and Fukai, 2018*) or implicitly (*Chenkov et al., 2017*) assumes the sequences produced by the network are initially induced in the network via modification of recurrent network weights, for example through homosynaptic mechanisms like spike-timing-dependent plasticity (STDP) (*Markram et al., 1997*; *Bi and Poo, 1998*; *Fiete et al., 2010*). While such models can indeed selectively bias network sequences, given the hypothesized weak magnitude of STDP, it is unclear whether such a mechanism could account for the awake trajectory replay observed in rodents, which can sometimes occur even when the animal has experienced the original trajectory only once (*Foster and Wilson, 2006*), nor whether it could be used to immediately induce novel computations that the animal cannot afford to learn over extensive repetitions. In our model we rely on a strong, fast-acting mechanism, LTP-IE, which although heterosynaptic (unlike STDP, LTP-IE follows spiking regardless of which input elicited the spikes [*Hyun et al., 2013*; *Hyun et al., 2015*) suffices in biasing a network toward producing highly specific sequences: LTP-IE 'tags' cells in the recurrent network with a higher probability of spiking in response to input (*Figure 1*), and sequential ordering is reconstructed by the existing but unmodified recurrent connectivity. While recent research has begun to investigate how small, biophysically plausible recurrent weight changes might encode new memories (*Curto et al., 2012*; *Yger et al., 2015*), to our knowledge this has not been applied in the context of sequences. We would also like to note that experience-dependent short-term plasticity mechanisms, like lingering presynaptic calcium following spiking, can transiently make a neural ensemble more sensitive to subsequent inputs (*Mongillo et al., 2008*), suggesting such a mechanism might be able to play a similar role to excitability changes.

In real brain networks, excitability changes likely interact with recurrent synaptic mechanisms like STDP or synaptic long-term potentiation (LTP) to shape computation. For instance, it has been observed that changes in how dendrites integrate EPSPs to trigger spiking (effectively an excitability change since this would affect spiking even if EPSPs remained stable) can co-occur with canonical STDP or LTP (*Daoudal et al., 2002*; *Daoudal and Debanne, 2003*; *Campanac and Debanne, 2008*; *Debanne and Poo, 2010*). Excitability vs synaptic changes may also coordinate the transference of memories across timescales. For example, as observed in vivo and as predicted by our model, hippocampal sequences that replay during awake quiescence are significantly compressed relative to behavioral timescales, suggesting replayed spikes may occur within the appropriate time windows for slower-acting STDP (*Markram et al., 1997*; *Bi and Poo, 1998*; *Dan and Poo, 2004*). Indeed, it was shown that replaying place-tuned firing sequences in vitro that were previously recorded in vivo could induce long-term connectivity changes between cells with overlapping place fields (*Isaac et al., 2009*, *J. Neurosci.*). Thus, LTP-IE may serve as a temporary buffer for sequences that

could become embedded in the network as long-term memories through gradual synaptic changes triggered by repeated replay. Additionally, LTP-IE-based replay in our model occurred over a wide range of parameters. While our networks were spatially homogeneous for all parameter sets (i.e. connectivity depended only on *differences* between place fields), this suggests potential robustness to connectivity heterogeneities within a single network, for example encoding preferred paths through an environment learned over longer timescales. Intuitively, paths defined by pre-existing stronger connections should bias replay in their favor but might also be overwritten by transient excitability changes encoding paths just traveled. Overall, short-term memory and computational flexibility and their subsequent effect on memory consolidation and long-term network structure likely rely on a host of dynamics and plasticity mechanisms, but we propose LTP-IE may play a significant role.

Finally, selectively biasing cell recruitment without changing recurrence generalizes beyond increasing spike responses through LTP-IE. An alternative mechanism is to simply send stronger baseline inputs to a subset of neurons, corresponding in our model to increasing the gate input rate $r^G$ for certain cells but not others, yet without applying LTP-IE. Such differential baseline inputs can act as context signals, and if interacting multiplicatively with sensory signals can control the gain of the latter without affecting their tuning; by modulating the gain of different neurons in such a way, one can then selectively and transiently encode context-dependent sensorimotor mappings without modifying synaptic weights (*Salinas, 2004b*; *Salinas, 2004a*), similar to how one might do so with LTP-IE. Given the functional similarity of selective LTP-IE or differential gain-modulating inputs, one would expect them to support similarly flexible sequential dynamics and signal routing, depending on the pre-existing but fixed recurrence structure. Indeed, neurons in entorhinal cortex, which inspired the gating signals G in our network, have been shown to support history-dependent, graded persistent activity (*Egorov et al., 2002*), suggesting the recurrent CA3 network could be subject to differential baseline inputs maintained over short-term memory timescales. Such a mechanism could act in concert with LTP-IE and could potentially act as an additional rapid and controllable gating signal governing effective excitabilities of neurons in the recurrent network.

## Model limitations

One question we did not explicitly address is whether and how a decay timescale of LTP-IE would affect sequence storage and replay. Indeed, with no forgetting or selective LTP-IE activation, one would expect all neurons in our model to undergo LTP-IE after the animal had fully explored the environment, thus preventing sequential reactivation along specific trajectories through the network. While this problem would be partially solved by considering additional input dimensions as described above (i.e. so that even if the x-y space were fully explored, only a fraction of the x-y-θ space would have been explored), one might also imagine additional neuromodulatory inputs serving to control the strength of LTP-IE during exploration. Indeed, while (*Hyun et al., 2013*; *Hyun et al., 2015*) did not observe a return of activation-triggered LTP-IE to baseline during the course of their slice experiments, in vivo LTP-IE in CA3 might be modulated by additional upstream inputs. For example, the neurotransmitter acetylcholine is known to interact with the voltage-gated potassium channel Kv1.2 (*Hattan et al., 2002*), the ion channel hypothesized to underlie LTP-IE (*Hyun et al., 2015*), and hippocampus receives state-dependent cholinergic inputs from medial septum (*Yoder and Pang, 2005*; *Mamad et al., 2015*). Furthermore, memory-related excitability changes in dentate gyrus have been shown to decay after two hours (*Pignatelli et al., 2019*), suggesting similar decay mechanisms may exist in CA3. Thus, sufficient machinery for erasing or selectively modulating the overall effects of LTP-IE, for example as a function of arousal or motivation state, may converge in the brain region that has served as the inspiration for our model. Clinically, one would also expect regulation of the spatiotemporal extent of excitability changes to be crucial in preventing pathological dynamics, such as incomplete signal transmission that might lead to short-term memory deficits (*Vallar and Shallice, 2007*), or hyperexcitability without sufficient compensatory inhibition leading to seizure-like events, as exhibited in the 'blowup' regime of our spiking network model (*Figure 3*).

Additionally, while we have assumed the upstream 'gating' inputs in our model are stochastic with homogeneous firing rates, cortical activity likely contains higher order structure beyond mean firing rates, such as the grid-like representations of space observed in neural firing patterns in entorhinal cortex (*O'Keefe and Burgess, 2005*; *Yamamoto and Tonegawa, 2017*), the hippocampal

input that inspired our gating inputs G. Our model demonstrates that replay can occur absent additional mnemonic information, but one would imagine such additional information would only increase the capacity and robustness for memory storage and computational flexibility, since this would act as an additional memory buffer. Furthermore, as mentioned above, entorhinal neurons may have the capacity for history-dependent persistent activity persisting over the same timescales as LTP-IE (*Egorov et al., 2002*), potentially supporting even more flexible heterogeneous gating inputs to the recurrent network. Investigating how such mechanisms would compete or synergize with LTP-IE could be a fruitful line of future inquiry.

What factors missing from our model might set the duration of replay events in more realistic situations? In our model, replay durations were determined by the length of the trajectory through the environment (4 m total in our simulations) and the replay speed was determined by the network parameters. Indeed, while empirical replay events and their accompanying sharp-wave ripples have typically been observed to occur over approximately 100 ms (*Carr et al., 2011*), when animals are allowed to explore larger environments replay can last several hundred milliseconds (*Davidson et al., 2009*), suggesting replay durations might be largely determined by the size of the environment, as in our model. Parallel lines of evidence, however, suggest hippocampal sequence replay may be gated by particular phases of slow, theta oscillations (4–8 Hz), which would constrain replay to occur within approximately 100–200 ms time windows (*Lisman and Idiart, 1995*; *Fuentemilla et al., 2010*; *Lisman, 2010*). While we excluded additional oscillatory components from our model for the sake of parsimony, one can imagine imposing time-dependent gating input rates $r^G(t)$ that, while spatially homogeneous, would oscillate at theta-like frequencies. One would expect such gating oscillations to constrain replay events to explicit phases of the theta cycle, thus preventing sequences of arbitrary duration from replaying.

Finally, an intriguing line of research suggests spatiotemporal propagation of activity in certain neural networks may follow a power-law distribution (*Beggs and Plenz, 2003*; *Beggs and Plenz, 2004*; *Plenz and Thiagarajan, 2007*; *Fiete et al., 2010*; *Scarpetta and de Candia, 2013*). Activity patterns distributed as such would imply scale invariance (scale-freeness) of activity structures and suggest the network state sits at a phase transition between dynamical regimes that may be optimal for processing inputs that vary over a large range (*Shew et al., 2009*). Since activity levels and durations in our network were capped by inhibition and refractoriness, and by the limited environment size, we would not expect our network to exhibit scale-free activity patterns; however, exploring what network mechanisms and connectivity statistics could yield power-law distributed sequences through LTP-IE and examining how these might be involved in sequence-based stimulus processing would be an exciting line of future inquiry.

## Experimental predictions

Despite its parsimony, our model of LTP-IE-based network sequences makes specific predictions that could be tested in hippocampal replay experiments. First, pharmacological blocking of LTP-IE during exploration should prevent the encoding of new trajectory information into CA3, consequently preventing subsequent replay. This could be achieved by in vivo application of nimodipine or PP2, which block intracellular calcium increase, protein tyrosine kinase activation, and Kv1.2 channel endocytosis, all potential mechanisms of LTP-IE, and which have been shown to block LTP-IE in vitro (*Hyun et al., 2015*). Second, inactivating medial entorhinal layer II (MEC II), which projects to CA3 and which we associate with gating inputs G in our analogy with hippocampal circuitry, should inhibit CA3 sequential replay events. Indeed, inactivation of MEC III inputs has been shown to inhibit forms of replay in CA1 (*Yamamoto and Tonegawa, 2017*); we propose that a parallel effect would be seen in MEC II inputs to CA3. Finally, the core prediction of our sequence induction model is that the neurons do not have to be initially activated in the order (or the reverse order) in which they will later reactivate. As the only information explicitly added to the network at the time of encoding is the *set* of neurons activated, speed or ordering information must be reconstructed from existing connectivity. While identifying the position tunings of individual neurons in vivo and subsequently artificially activating them out of order currently presents a substantial technical challenge, one could potentially achieve a similar effect using a virtual reality environment with place-specific sensory cues, and in which specific places along a virtual trajectory were presented out of order. Our model predicts that as long as the animal had sufficient knowledge of which sensory cues corresponded to

which environment locations, replay would occur in an order corresponding to a path through the environment, rather than the order in which the virtual places were presented.

## Computational significance

Besides revealing a potential core mechanism for sequence induction, our model lends two key insights into the structure and control of replay in biological neural networks. First, it suggests a link between isotropy in pre-existing spatial representations in the network (i.e. the spatially organized connectivity) and the experimental observation that replay sometimes occurs in the forward, and sometimes in the reverse order relative to the neurons' initial activation during the trajectory (*Foster and Wilson, 2006*; *Davidson et al., 2009*; *Carr et al., 2011*). In particular, since no directionality information is stored by LTP-IE (which only tags which neurons were recently active), replay should occur with equal chance in forward or reverse. (Note however, that as observed in *Davidson et al., 2009*, replay events may have an increased probability of originating at cells tuned to the animal's current location, as a result of place-tuned inputs to the network during awake quiescence.) One would therefore not expect to observe reverse replay for neural sequences that do not have a natural spatial embedding, for example spoken sentences in humans. Second, the upstream gating signal in our model might serve as a substrate for the short-term state-dependence of replay, for example with arousal or motivation controlling when the gating signal is present or absent. This could ensure that replay occurs only at appropriate times (e.g. quiescent rest) without interfering with other computations being performed by the network, such as pattern separation in external sensory inputs (*Leutgeb et al., 2007*; *Bakker et al., 2008*).

Further, while our spiking network was embedded in 2-D space, our results frame the more general question of how a network embedded on a higher dimensional manifold, or with no natural embedding, supports excitability-based sequence induction and information flow. For example, one could imagine the rapid induction of spike sequences corresponding to movement along 3-D trajectories if the recurrent architecture reflected a 3-D space, as might occur within the 3-D hippocampal place-cell network of bats (*Yartsev and Ulanovsky, 2013*). More generally it remains an open and intriguing question as to which network structures are most suited to rapid excitability-based flexibility among sequences and behaviorally relevant computations. One example of a network model with no obvious spatial embedding, and which could potentially be recast in the framework of excitability-based information flow, is the 'binding pool' model of associative short-term memory proposed by *Swan and Wyble (2014)*. Here, associations are encoded in a set of steady, persistently active 'binding' units that allow preferential signal transmission among specific sets of units representing object features, to which the binding units are randomly connected. While the role of such persistent spiking in coding mnemonic information is presently undergoing substantial reevaluation (*Sreenivasan et al., 2014*), one could theoretically replace it with excitability increases, with an equivalent consequence for information transmission. We explored this possibility in the context of LTP-IE using a reduced model of LTP-IE (*Figure 5D,E*), in which we were able to encode stimulus-response mappings in a random network by applying LTP-IE to a specific set of 'switchboard' neurons (analogous to *Swan and Wyble, 2014*). Notably, since connectivity here required no training, this architecture may be well suited to transiently storing novel mappings through LTP-IE that the network has never seen before, so long as the neural ensembles encoding the relevant stimuli and responses were sufficiently (but randomly) connected to the switchboard/binding pool. The nuances of how such a network might implemented in biological spiking networks remains an open question, but our present work suggests its theoretical plausibility.

Finally, one can draw an equivalence between the modulation of cellular properties to shape sequences and information flow and selective variation in upstream modulatory inputs to a cell. For example, a neuron receiving strong inhibition might be less likely to participate in a computation than one receiving excitation, since the latter would be more likely to spike in response to inputs, similar to if one had increased its excitability through intrinsic cellular mechanisms. Indeed, it was recently shown that external modulatory inputs to a recurrent network could control the speed at which a computation performed by the network unfolded through time, closely recapitulating experimental results (*Remington et al., 2018*) More flexible control of neural information transmission via modulatory inputs has also been explored in the framework of gain-modulating context inputs that can effectively move a network into different stimulus-response mappings, also without changing synaptic connectivity (*Salinas, 2004b*; *Salinas, 2004a*). We propose that the control of information

flow and computation in recurrent networks via effective excitability alone, without structural connectivity changes, might be a rich framework for investigating rapid cognitive flexibility and short-term information storage. This may lead to significant insights into how we adapt our mental processes and behaviors in the face of ever-changing environments and task demands.

# Materials and methods

## Membrane and synaptic dynamics for spiking network simulation

We modeled single neurons as leaky integrate-and-fire neurons with conductance-based synapses according to the equation

$$\tau_m \frac{dV^i}{dt} = -(V^i - E_L) - g_E^i(t)(V^i - E_E) - g_I^i(t)(V^i - E_I) + I_{ext}^i(t)$$

where $V^i$ is the membrane voltage of neuron $i$; $\tau_m$ is the membrane time constant (50 ms for pyramidal cells [PCs], 5 ms for inhibitory cells [INHs]); $E_{leak}$ is the leak/resting potential (−68 mV for PCs, −60 mV for INHs); $g_E^i$ and $g_I^i$ are the time-varying conductances of excitatory and inhibitory synaptic inputs, respectively; $E_E$ = 0 mV and $E_I$ = −80 mV are the excitatory and inhibitory synaptic reversal potentials, respectively; and $I_{ext}^i$ is any externally injected current to cell $i$ (in units of the expected deflection in membrane voltage). If $V^i$ exceeded a threshold $v_{th}$ = −36 mV (or $v_{th}$ = −50 mV for INHs) at any time step, a spike was emitted and $V^i$ was reset to $E_{leak}$ for a refractory period $\tau_r$ 8 ms (or $\tau_r$ = 2 ms for INHs). The outsize PC refractory period ensured that the sequential events in our simulations propagated only in one direction and is consistent with refractory periods measured in hippocampal region CA3 during some experiments (*Raastad, 2003*).

Synaptic dynamics were conductance-based and were modeled as exponentially filtered trains of weighted delta functions representing input spikes arriving from different upstream neurons. Specifically, for an individual neuron $i$

$$g_E^i(t) = \sum_j \sum_k w_{ij} h(t) * \delta(t - t_k^j)$$

and similarly for $g_I^i$; here $j$ indexes neurons and $k$ indexes spikes, such that $w_{ij}$ is the synaptic weight from neuron $j$ onto neuron $i$ and $t_k^j$ is the $k$-th spike of the $j$-th neuron; and h(t) was a one-sided exponential with time constant $\tau_E$ 2 ms. In our simulations we assumed all synapses were excitatory and did not explicitly model effects of inhibition. Synaptic weight values are unitless and specify the postsynaptic conductance change in response to a presynaptic spike, relative to the leak conductance.

## Pyramidal cell tuning parameters

In network simulations, we assigned each PC a position (x$^i$, y$^i$) corresponding to the peak of its tuning curve, that is the (x, y) position eliciting maximal firing. Positions were assigned to approximately tile a grid spanning −1 to 1 m in both dimensions (*Figures 2* and *3*) or −2 to 2 m (*Figure 4*), using 3000 or 12000 PCs, respectively. We used the following position-dependent firing-rate equation to calculate the maximal firing rate we expected to be evoked by the trajectory through the simulated environment (the firing rate evoked by the closest point on the trajectory to the neuron's place field center):

$$r^i(x, y) = r_{max} \exp[-((x - x^i)^2 + (y - y^i)^2)/(2\lambda_{PL}^2)]$$

where $r_{max}$ = 20 Hz is the maximum firing rate (i.e. when the trajectory passes directly through the neuron's peak tuning position) and $\lambda_{PL}$ PL0.15 m is the length constant determining how close a neuron's peak tuning must be to the trajectory to significantly activate. All neurons had the same $r_{max}$ and $\lambda_{PL}$.

## Spiking network architecture

We included an excitatory (PC [pyramidal cell]) and inhibitory (INH) population in our spiking network, as well as random spiking inputs from a gating signal G impinging on the PC population. We

included excitatory recurrent connections among PCs, connections from PCs to INHs and from INHs to PCs, and only PCs received G inputs. Values for synaptic weights are unitless and specify the post-synaptic conductance change caused by a single presynaptic spike, relative to the leak conductance.

### Recurrent excitatory connectivity

Recurrent excitatory synaptic weights were assigned such that neurons with similar position tuning had stronger connectivity, with the connectivity between two neurons $i$ and $j$ with peak tunings separated by $d_{ij}$ given by

$$w_{ij} = w^{PC,PC}\exp[-d_{ij}^2/(2(\lambda^{PC,PC})^2)].$$

This was motivated by past modeling studies (**Káli and Dayan, 2000**; **Solstad et al., 2014**) and by analyses of correlated hippocampal activity in the absence of sensory input (**Giusti et al., 2015**). Note that in our network architecture, all recurrent connections between position-tuned cells are reciprocal, due to the symmetry of the distance function $d_{ij}$. All $w_{ij}$ below a minimum value $w_{min}^{PC,PC} = 0.1$ were set to 0.

### Inhibitory connectivity

We included an INH population 1/10 the size of the PC population (300 for **Figures 2–3** and 100 for **Figure 4**). Each PC projected to each INH with probability 0.5 and weight $w^{INH,PC} = 0.03$. For **Figures 2–3**, each INH projected to each PC with probability 0.5 and weight $w^{PC,INH} = 0.02$ (except where $w^{PC,INH}$ was varied in the parameter sweeps in **Figure 3**). For **Figure 4** we decreased $w^{PC,INH}$ to 0.002 to account for the larger inhibitory pool.

### Gating input connectivity

Synaptic weights $w_i^G$ on gating inputs G were initially uniform across all position-tuned neurons, and each neuron received an independent instantiation of an upstream G spike train, generated from a Poisson-distributed point process with constant rate $r^G$. While effective $w_i^G$ varied as a function of each neuron's trajectory-dependent LTP-IE, $r^G$ was identical across all neurons regardless of tuning or LTP-IE status.

## Long-term potentiation of intrinsic excitability (LTP-IE) of gating inputs

Although LTP-IE is thought to result from changes in dendritic conductances (**Hyun et al., 2015**), we modeled it as an effective synaptic weight change (since conductance changes can be absorbed into connection weights according to the equations underlying conductance-based synaptic dynamics), such that initial weights $w_i^G$ were scaled by a factor $\sigma_i$ as a function of neuron $i$'s position along the initial trajectory. For computational efficiency we did not model network activity during the initial trajectory, but instead directly calculated the $\sigma_i$ expected to result from the maximum firing rate $r_{max}^i = \max(r^i(x, y) \mid (x, y) \in \text{trajectory})$. From $r_{max}^i$ we computed $\sigma_i$ according to the following equation (**Figure 1B**):

$$\sigma_i = 1 + \frac{\sigma_{max} - 1}{1 + \exp[-\beta_\sigma(r_{max}^i - r_\sigma)]}$$

with $r_\sigma$=10 Hz and $\beta_\sigma$σ1. Thus, neurons with position-tuning peaks close to the original trajectory had a σ$_i$ near σ$_{max}$ whereas neurons far away from the trajectory had σ$_i$ near unity.

## Sequence replay

We fully simulated the replay epoch only, during which all neurons in the recurrent network received independent stochastic gating input G at constant rate $r^G$. Replay events arose spontaneously, but in **Figure 4** we triggered sequence propagation with a short current pulse to neurons in one of the sensory regions; trigger parameters were unimportant as long as they elicited significant spiking activity in the neurons at the start of the sequence. In **Figure 2—figure supplement 1F**, replayed positions were decoded during by taking the median peak-tuning of neurons that spiked within 5 ms windows spanning the replay event (decoding was only performed in a window if there were at least five spikes in that window).

### Replay event detection and speed measurements

To detect spontaneous replay events automatically, we first smoothed the cell-averaged PC firing rate with a Gaussian filter with a 2 ms standard deviation. Next, all timepoints where this smoothed firing rate exceeded 0.5 Hz were marked. Continuous events above 0.5 Hz lasting longer than 30 ms were then labeled as replay events, with two replay events joined into one if the duration of the gap between them was less than 10 ms.

We measured replay speed for an event by first computing the linear position of each PC's place field center along the Z-shaped trajectory shown in *Figure 2E*. We then plotted these linear positions of all PCs that spiked during the middle 80% of the replay epoch vs the times the spikes occurred (we removed the first and last 10% to avoid edge effects). Finally, we calculated speed by computing the slope of the best-fit linear regression (minimal least squared error) to the linear-position-vs-spike-time plot.

### Spiking network simulations

#### Single-neuron LTP-IE simulation (*Figure 1*)

LTP-IE-level-dependent voltage distributions (*Figure 1D,E*) were computed using 15 s simulations. In measuring *Figure 1F–G*, current pulses were 10 ms in duration, and spiking probability was measured within this window. Current-evoked spike probabilities (*Figure 1G*) were computed using 125 s simulations, with pulses presented every 250 ms.

#### Replay in networks of neurons (*Figures 2–3*)

In replay simulations we used 3000 neurons, with place fields distributed on an approximate lattice over a 2 m x 2 m environment. Gating inputs were provided continually, as our simulation represented the awake, quiescent state during which replay is thought to occur. Gating inputs were random homogeneous Poisson point processes, independent to each neuron, with rate $r^G$ that was the same across all neurons.

#### Replay dynamics parameter sweep (*Figure 3*)

For each parameter set we ran ten 10 s simulations starting with different random number generator seeds. We calculated spontaneous event frequency by counting the number of spontaneous replay events arising during each simulation, dividing by the 10 s simulation duration, and averaging over the 10 trials.

To explore the dependence of spontaneous replay event frequency on our network parameters, we first varied the magnitude factor $w^{PC,PC}$ of the recurrent excitatory weight profile and the inhibitory-to-PC connectivity strength $w^{PC,INH}$ (*Figure 3A*), while holding all other parameters fixed. We next varied $w^{PC,PC}$ and $\lambda^{PC,PC}$, the recurrent excitatory connectivity length scale, respectively, while holding all other parameters fixed (*Figure 3B*). Finally, we varied $\sigma_{max}$, the maximum LTP-IE level, and $r^G$, the gate-input rate, while holding all other parameters fixed (*Figure 3C*). Fixed parameter values are given in the table below.

#### Virtual replay speed measurements and parameter sweep

To explore the dependence of virtual replay speed on network parameters we varied $w^{PC,PC}$ and $\lambda^{PC,PC}$ while measuring replay speed in the following way. For all parameter sets in *Figure 3A* that did not exhibit 'blowup' behavior (i.e. all parameter sets where replay events were roughly confined to the LTP-IE profile), we ran up to twenty 600 ms trial simulations in which we injected a short current pulse to the initial end of the trajectory to trigger replay. Simulations ended if 20 trials were reached or 10 successfully triggered replay events were observed. To calculate the virtual replay in a given event we first truncated the initial and final 10% of the event to avoid edge effects. We then collected all neurons that spiked during the event and fit a line using least-squares regression to the linear positions of preferred (x, y) of each neuron along the trajectory vs the times those neurons emitted spikes, and the slope of this line was taken to be the virtual replay speed. We only included replay events lasting between 50 and 400 ms for the speed computation, and events exhibiting a replay speed of more than 10 m/s (events falling outside this range tended not to correspond to unidirectional propagation of the spike sequence along the LTP-IE-defined trajectory). To make the

plot shown in **Figure 3G** we computed the average virtual replay speeds for all parameter sets exhibiting at least five successful replay events.

## Power spectral density analysis

Power spectral densities shown in **Figure 3H** were computed by running a single simulation for each of two different parameter sets, collecting all replay events, measuring the power spectrum of each event, and averaging over events.

## LTP-IE-based stimulus-response mappings (**Figure 4**)

For this simulation, in order to construct sufficiently non-interfering LTP-IE-defined paths, we constructed a model network of 12000 neurons with neural place fields ranging from −2 to +2 m in both the x- and y- dimensions and let two clusters of radius of 0.25 m, centered at (−1, 1) and (−1,−1), represent two stimuli S1 and S2, respectively, and two additional clusters, centered at (1, 1) and (1, -1), encode motor outputs M1 and M2, respectively. Readout units summed activity in any of the four clusters at every timestep. Sequence reactivation was triggered by depolarizing current injections into cells contained in the S1 or S2 clusters.

## All parameters for spiking network simulations

We used the following values for all simulation parameters unless otherwise noted (e.g. when values were varied in parameter sweeps [**Figure 3**]):

| Symbol | Definition | Value | Symbol | Definition | Value |
|---|---|---|---|---|---|
| $\tau_m^{PC}$ | PC membrane time constant | 50 ms | $N^{PC}$ | Number of PCs in recurrent network | 3000 (**Figures 2–3**) 12000 (**Figure 4**) |
| $E^{PC}_{leak}$ | PC leak potential | −68 mV | $N^{INH}$ | Number of INHs in recurrent network | 300 (**Figures 2–3**) 1200 (**Figure 4**) |
| $v^{PC}_{th}$ | PC spike threshold | −36 mV | $w^{PC,G}$ | Gating input weight without LTP-IE | 0.8216 (**Figures 2–3**) 0.5 (**Figure 4**) |
| $\tau^{PC}_r$ | PC refractory period | 8 ms | $w^{PC,PC}$ | Excitatory recurrent weight scale factor | 2.6 |
| $\tau_m^{INH}$ | INH membrane time constant | 5 ms | $w_{min}^{PC,PC}$ | Min nonzero recurrent weight | .1 |
| $E^{PC}_{leak}$ | INH leak potential | −60 mV | $w^{PC,INH}$ | INH→PC connection weight | 0.02 (**Figures 2–3**) 0.002 (**Figure 4**) |
| $v^{PC}_{th}$ | INH spike threshold | −50 mV | $w^{INH,PC}$ | PC→INH connection weight | .03 |
| $\tau^{PC}_r$ | INH refractory period | 2 ms | $p^{PC,INH}$ | INH→PC connection probability | 0.5 |
| $E_E$ | Excitatory synaptic reversal potential | 0 mV | $p^{INH,PC}$ | PC→INH connection probability | 0.5 |
| $\tau_E$ | Excitatory synaptic time constant | 2 ms | $r^G$ | Gating input firing rate | 125 Hz (**Figures 2–3**) 200 Hz (**Figure 4**) |
| $E_I$ | Inhibitory synaptic reversal potential | −80 mV | $\lambda^{PC,PC}$ | Excitatory recurrent connectivity length scale | .053 m |
| $\tau_I$ | Inhibitory synaptic time constant | 2 ms | $\sigma_{max}$ | Max LTP-IE value | 2 (**Figures 2–3**) 1.84 (**Figure 4**) |
| $r_{max}$ | Max position-driven firing rate | 20 Hz | $r_\sigma$ | Threshold firing rate for LTP-IE | 10 Hz |
| $\lambda_{PL}$ | Position-tuning length constant | 0.15 m | $\beta_\sigma$ | Scale factor for LTP-IE onset | 1 |
| | | | $\Delta T$ | Simulation time step | 0.5 ms |

## Reduced model dynamics

We used a reduced network model and a reduced model of LTP-IE to explore the algorithmic potential of LTP-IE separately from its specific biophysical properties.

The reduced network model operates in discrete timesteps, with each neuron's 'voltage' $v^i$ at time $t$ given by the sum of all of its inputs:

$$v^i(t) = \sum_j w_{ij} s_j(t-1) + i_{ext}(t) + i_g \sigma^i$$

where $w_{ij}$ is the synaptic connection weight from neuron $j$ onto neuron $i$, $s_j(t-1)$ is one if neuron $j$ spiked at $t - 1$ and 0 otherwise, $i_{ext}(t)$ is the external input current to neuron $i$ at time $t$, $i_g$ is a constant 'gating' current, and $\sigma^i$ is neuron $i$'s LTP-IE level (ranging from 1 to 2, depending on the neuron). If a neuron's voltage at time $t$ exceeded a threshold voltage $v_{th}$, we let that neuron spike, with the following caveat: a maximum of $n_{max}$ neurons were allowed to spike at any individual timestep, and if more than $n_{max}$ neurons' voltages exceeded $v_{th}$, we let only the $n_{max}$ neurons with the highest voltages spike. Neurons that spiked subsequently entered a refractory period that prevented them from spiking for $\tau_r$ subsequent timesteps. LTP-IE values $\sigma^i$ were static throughout each simulation and pre-assigned according to rules that depended on the specific simulation in question.

## Reduced model simulation of intersecting trajectory replay

### Connectivity

We first specified tuning parameters for each neuron with regard to position and orientation in the environment. Each neuron was assigned a position preference in (x, y) space, tiling a (2 m x 2 m) environment; as well as a preferred θ, randomly selected from one of 8 possible angles tiling 2π radians: 0, π/4, -π/4, ..., such that each window of 8 neurons in a set of windows tiling a row of neurons contained all eight angles. (E.g. the first through eighth neurons with a given preferred y would each have a distinct preferred θ, similarly for the ninth through 16th neurons, and so on.) Neurons were subsequently connected according to their preferred tuning, with the (symmetric) connection weight $w$ between two neurons separated by Δx, Δy, and Δθ in their preferred tuning space given by

$$w(\Delta x, \Delta y, \Delta \theta) = \exp\left(-\frac{\Delta x^2 + \Delta y^2}{2\lambda_{xy}^2}\right) \exp\left(-\frac{\Delta \theta^2}{2\lambda_\theta^2}\right)$$

with $\lambda_{xy}$ and $\lambda_\theta$ specifying connectivity length constants. (Note Δθ is the absolute distance between two angles calculated on the unit circle.)

### LTP-IE profile

To assign LTP-IE levels to neurons such that accurate replay would be achieved, we first generated the trajectory through the environment shown in *Figure 5A*, inset. Neurons with preferred tuning close to the trajectory in (x, y, θ)-space were assigned high LTP-IE levels (σ ~2), and neurons far away from the trajectory were assigned low LTP-IE levels (σ ~1), according to the following equation:

$$\sigma(x, y, \theta) = 1 + \frac{1}{1 + \exp[-\beta(z - z_0)]}$$

$$z(x, y, \theta) = \exp\left[-.5\left(\frac{\Delta x^2}{\sigma_x^2} + \frac{\Delta y^2}{\sigma_y^2} + \frac{\Delta \theta^2}{\sigma_\theta^2}\right)\right]$$

where Δx, Δy, and Δθ in this case were the distance between a neuron's preferred x, y, and θ, respectively, and the point on the trajectory that yielded the highest z. $\sigma_x$, $\sigma_y$, and $\sigma_\theta$ were length constants determining how similar a neurons tuning peak had to be to an (x, y, θ) point on the trajectory to induce LTP-IE.

### Trigger

In the reduced model we explicitly triggered replay by injecting an external current pulse with a squared exponential profile focused at either the initial (−1 m, 0.25 m) or final end (−0.25 m, 1 m) to

evoke forward or reverse replay, respectively. Exact trigger parameters were unimportant as long as they evoked spiking in approximately the correct location.

## Parameters
We used the following parameter values for the reduced model simulation of intersecting trajectory replay:

| Symbol | Definition | Value | Symbol | Definition | Value |
|---|---|---|---|---|---|
| $N$ | Total number of neurons | 4096 (64 × 64) | $n_{max}$ | Max number of active neurons at a timestep | 10 |
| $N^X$ | Number of neurons in each row of environment | 64 | $\lambda_{xy}$ | Connectivity length scale | 0.25 m |
| $N^Y$ | Number of neurons in each column of environment | 64 | $\lambda_\theta$ | Connectivity orientation-length scale | $\pi/7$ radians |
| $N^\theta$ | Number of equally spaced orientations | 8 | $\sigma_X, \sigma_Y$ | LTP-IE length scales | 0.2 m |
| $v_{th}$ | Spike threshold | 11 | $\sigma_\theta$ | LTP-IE orientation-length scale | $\pi/16$ radians |
| $i_G$ | Gate input | 5 | $\beta$ | LTP-IE sigmoid steepness | 20 |
| $\tau_r$ | Refractory period | six timesteps | $T$ | Number of timesteps in simulation | 50 timesteps |

## Reduced model simulation of non-spatial mapping

### Connectivity
We included five groups of neurons in this simulation: S1, S2, M1, M2, and B. S1 and S2 represented sensory ensembles, each composed of $N_S$ neurons; and M1 and M2 represented motor ensembles, each composed of $N_M$ neurons. B represented a 'switchboard' ensembles of neurons, akin to Swan and Wyble's 'binding pool' (*Swan and Wyble, 2014*), comprising $N_B$ neurons. Each motor ensemble, with connection weight $w^{MM}$.

$N^{BS}$ randomly selected switchboard received connections from S1, and another potentially overlapping $N^{BS}$ randomly selected switchboard neurons received connections from S2. If a switchboard neuron was included in the group receiving connections from S1, $n^{BS}$ neurons from S1 were selected to send connections to that switchboard neuron; similarly for S2. Sensory-to-switchboard connections had synaptic weights $w^{BS}$.

$N^{MB}$ randomly selected switchboard neurons sent connections to M1, and another potentially overlapping $N^{MB}$ randomly selected switchboard neurons sent connections to M2. If a switchboard neuron was included in the group sending connections to M1, $n^{MB}$ neurons in M1 were selected to receive connections from that switchboard neuron; similarly for M2. Switchboard-to-motor connections had synaptic weight $w^{MB}$.

There were no recurrent connections within either sensory ensemble or within the switchboard ensemble. These connections were omitted since our goal was to demonstrate a proof-of-concept nonspatial-mapping computation through LTP-IE. Exploring the effects of sensory and switchboard recurrence in a more biophysically accurate model may yield fruitful insights into LTP-IE-based computations, however.

### LTP-IE profile
To encode the mapping (S1→M1, S2→M2) we assigned an LTP-IE value of σ = 2 to all switchboard neurons that either (1) received connections from S1 and sent connections to M1 or (2) received connections from S2 and sent connections to M2. All other neurons were assigned an LTP-IE value of σ = 1. To encode the mapping (S1→M2, S2→M1) we assigned an LTP-IE value of σ = 2 to all switchboard neurons that either (1) received connections from S1 and sent connections to M2 or (2) received connections from S2 and sent connections to M1. All other neurons were assigned an LTP-IE value of σ = 1.

### Trigger
We triggered recall of an association encoded in the switchboard LTP-IE profile by injecting an external current pulse at timestep two into all the neurons in either S1 or all the neurons in S2.

## Parameters

| Symbol | Definition | Value | Symbol | Definition | Value |
|---|---|---|---|---|---|
| $N$ | Total number of neurons | 2400 | $N^{BS}$ | Number of switchboard neurons receiving sensory projections | 200 |
| $N_S$ | Number of neurons in each sensory ensemble | 100 | $N^{MB}$ | Number of switchboard neurons projecting to one motor ensemble | 200 |
| $N_M$ | Number of neurons in each motor ensemble | 100 | $n^{BS}$ | Number of sensory neurons projecting to each switchboard neuron receiving sensory input | 10 |
| $N_B$ | Number of neurons in switchboard ensemble | 2000 | $n^{MB}$ | Number of motor neurons receiving projections from each switchboard neuron projecting to motor ensembles | 25 |
| $v_{th}$ | Spike threshold | 10 | $w^{MM}$ | Recurrent connection strength within each motor ensemble | 0.35 |
| $i_G$ | Gate input | 3 | $w^{BS}$ | Connection strength from sensory ensemble to switchboard neurons | 0.88 |
| $\tau_r$ | Refractory period | five timesteps | $w^{MB}$ | Connection strength from switchboard to motor ensemble neurons | 2.33 |
| $n_{max}$ | Max number of neurons active at a timestep | 50 | $T$ | Number of timesteps in simulation | five timesteps |

## Code

All code for this work was written in Python three and is available at https://github.com/rkp8000/seq_speak. (*Pang, 2019*; copy archived at https://github.com/elifesciences-publications/seq_speak).

## Acknowledgements

We would like to acknowledge Stefano Recanatesi, Alison Duffy, Guillaume Lajoie, Ben Lansdell, Yoni Browning, and Kenneth Latimer for helpful discussions regarding this work. We would also like to acknowledge the MONA2: Modeling Neuronal Activity meeting for providing an opportunity for feedback on an earlier version of this work.

## Additional information

### Funding

| Funder | Grant reference number | Author |
|---|---|---|
| Simons Foundation | Collaboration for the Global Brain | Adrienne L Fairhall |
| University of Washington | Computational Neuroscience Training Grant | Rich Pang |
| Washington Research Foundation | UW Institute for Neuroengineering | Adrienne L Fairhall |
| National Institutes of Health | R01NS104925 | Adrienne L Fairhall |

The funders had no role in study design, data collection and interpretation, or the decision to submit the work for publication.

## Author contributions
Rich Pang, Conceptualization, Resources, Software, Formal analysis, Supervision, Validation, Investigation, Visualization, Methodology, Writing—original draft, Project administration, Writing—review and editing; Adrienne L Fairhall, Supervision, Funding acquisition, Writing—review and editing

## Author ORCIDs
Rich Pang (iD) https://orcid.org/0000-0002-2644-6110

## Decision letter and Author response
Decision letter https://doi.org/10.7554/eLife.44324.011
Author response https://doi.org/10.7554/eLife.44324.012

# Additional files

## Supplementary files
• Transparent reporting form
DOI: https://doi.org/10.7554/eLife.44324.009

## Data availability
This paper is a modelling study that did not generate any new data or analyse any previous datasets. Code used for modelling is available on GitHub at https://github.com/rkp8000/seq_speak (copy archived at https://github.com/elifesciences-publications/seq_speak).

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
