## [Decision Letter]

Thank you for submitting your article "Fast and flexible sequence induction in spiking neural networks via rapid excitability changes" for consideration by *eLife*. Your article has been reviewed by three peer reviewers, including Emilio Salinas as the Reviewing Editor and Reviewer #1, and the evaluation has been overseen by Eve Marder as the Senior Editor.

The reviewers have discussed the reviews with one another and the Reviewing Editor has drafted this decision to help you prepare a revised submission.

Summary:

This manuscript investigates, via computer simulations, the functional consequences of a physiological phenomenon known as long-term potentiation of intrinsic excitability (LTP-IE), whereby neurons that fire intensely (due to input S) for a short period of time then respond more strongly to subsequent input (from a different source G). This mechanism is important for several reasons: 1) reports of LTP-IE show a two-fold magnitude increase in EPSPs, substantially higher than what is reported for STDP, and 2) as the authors demonstrate, it can quickly generate highly effective, transient communication channels that may mediate cognitive flexibility. The computer simulations show that this mechanism can explain the replay of place cell responses observed in the hippocampus; specifically, the fact that such activation sequences can replay in forward or reverse order, and over a compressed time scale. More generally, this mechanism is also shown to allow the reactivation of arbitrary response sequences for cells embedded within a recurrent network.

The introduction of intrinsic plasticity concepts to the analysis of population sequences is highly innovative and quite interesting. Although the model has limitations, it represents a feasible and simple mechanism that could underlie hippocampal replay events, and it has other, more general implications. The manuscript is interesting, even provocative, and will certainly enrich the field.

Essential revisions:

The paper requires some updates with regard to the discussion of previous findings by other labs, and some clarification to maximize its impact.

1) Is LTP-IE-time-restricted in the model, or is it assumed to be long-lasting? This is an important question for two reasons. First, because of intersecting trajectories, which could substantially limit the proposed model. The model proficiently generates simple trajectories, such as ABCDE…, but in naturalistic environments (such as open field) there is replay of complex sequences, such as ABCDAEFG. Second, a similar issue arises when the trajectory inducing LTP-IE is very long, as this model would facilitate replays of any length.

The authors should consider some evidence showing that LTP-IE might be time-limited, see e.g. Pignatelli et al., 2019, which would reduce the impact of both of these concerns.

2) Explaining the replay of complex sequences, as mentioned above, would be a major contribution to the field, but it is not clear whether the model can generate those or not. There are many proposed models of reactivation of simple sequences, but none (as far as I know) for reactivation of complex/realistic sequences. While the authors are commended for proposing a new biophysical mechanism for explaining replay, this a moderate contribution to the field. What would be really exciting would be a model such as the one they propose that goes beyond the kind of simple sequences typically found in hippocampus studies with rodents running on linear tracks.

3) Discussion, last paragraph and elsewhere: the authors clearly state that their model works in the absence of synaptic modification, in fact this is one of the major points. This idea (that intrinsic plasticity can act based on existing, but not changing synaptic connectivity) has been presented by other groups before and this prior work should be cited and discussed in appropriate detail.

For instance, previous modeling work (Salinas, 2004; Salinas, 2004) showed that changes in excitability very similar to those proposed here, in which one input modulates the response to another, are ideal for switching a network from one functional configuration (say, sensory-motor map 1) to another (say, sensory-motor map 2). Although the changes in excitability in that earlier work went by a different name, "gain modulation", the underlying mathematics would still apply, so discussing those earlier findings would bolster the argument that LTP-IE could indeed be computationally effective for the sorts of remapping proposed in this manuscript.

The work also needs to be discussed in the context of Moshe Abeles' neuronal avalanche concept (despite its stronger focus on the neocortex).

In the context of LTP-IE, it might be interesting to discuss a recent finding that spike firing can be independent of dendritic EPSP amplitudes, and is instead intrinsically controlled, see Ohtsuki and Hansel, 2018.

4) The authors state that 'this mechanism….is observed in hippocampus only'. If this statement refers to LTP-IE, it is not correct, see e.g. Daoudal, Hanada and Debanne, 2002. Debanne's work should also be cited with regard to STDP. If the statement does not refer to LTP-IE, this needs to be stated more precisely.

5) Given the co-occurrence of replay and sharp-wave ripples, the lack of inhibition in the current model may be a substantial limitation. Replay was triggered using depolarizing current, however the physiological relevance of the model would be bolstered if replay could be triggered by a sharp-wave ripple event in an inhibitory population. More broadly, how does the model behave if one were to add inhibition in addition to excitation?

6) Related to the above point, it was difficult to evaluate the robustness of the model. There are many parameters, but which ones are more critical for the results, and what is the 'operating range' of the model? Is the model robust to small perturbations in parameters?

7) Regarding Figure 4, it is unclear where in the brain a recurrent sensorimotor network with LTP-IE might exist to support the generation of transient associations using the proposed mechanism. What do the authors suggest is the biophysical substrate for non-spatial transient associations?

8) Some parts of the manuscript were found to be somewhat technical and difficult to follow. Specifically, throughout the Results: second paragraph; subsection “LTPIE increases membrane voltages and spike rates under random gating inputs”, first and last paragraphs; subsection “Spike sequences propagate along LTP-IE-defined paths through a network”, second paragraph; subsection “Dependence of LTP-IE-based sequence propagation on network parameters”, first paragraph and subsection “LTP-IE-based sequences can encode temporary stimulus response mappings”, last paragraph. The Results should be more accessible to a broader readership not specialized in computational neuroscience.

For instance, it is not easy to understand how the results in Figure 2 relate to, and emerge from, those in Figure 1. In general, it would be useful if the authors could better explain the relationship between their neuronal mechanism and the results.

In addition, while the model works well for encoding spatial trajectories in which the closeness of individual locations is pre-encoded in the recurrent connectivity weights, it is difficult to see how this would extend to the given example of cognitive flexibility (i.e., raising our right hand when a particular word is heard, and the left hand when a different word is heard), particularly if no a priori spatial relationship exists. Are spatial relationships necessary? There is no evidence provided to suggest that the neocortical network has the necessary architecture to support LTP-IE based replay.

---

## [Author Response]

Essential revisions:The paper requires some updates with regard to the discussion of previous findings by other labs, and some clarification to maximize its impact.1) Is LTP-IE-time-restricted in the model, or is it assumed to be long-lasting? This is an important question for two reasons. First, because of intersecting trajectories, which could substantially limit the proposed model. The model proficiently generates simple trajectories, such as ABCDE…, but in naturalistic environments (such as open field) there is replay of complex sequences, such as ABCDAEFG. Second, a similar issue arises when the trajectory inducing LTP-IE is very long, as this model would facilitate replays of any length.The authors should consider some evidence showing that LTP-IE might be time-limited, see e.g. Pignatelli et al., 2019, which would reduce the impact of both of these concerns.

We agree that time-limiting the influence of LTP-IE would be crucial in physiological contexts. First, unchecked excitability increases across a whole network could lead to pathological dynamics like seizure-like events unless inhibition was also correspondingly strengthened. Second, if all neurons underwent LTP-IE, e.g. through a long trajectory that covered all neurons’ place fields in the network, then the memory of the trajectory would be lost since all cells would be potentiated.

Since our simulations lasted tens of seconds at most, presumably less than a potential LTP-IE decay timescale, we did not explicitly model LTP-IE decay in our revised simulations. However, we have addressed potential mechanisms and consequences of such decay in the first paragraph of the “Model limitations**”** section of our Discussion, and we have referenced the work of Pignatelli et al., who found that excitability changes in the dentate gyrus decayed over about two hours.

Regarding the replay of complex sequences, we did not attempt to model this in our spiking network due to the complexity involved in identifying a valid parameter regime. However, we have addressed the problem of complex sequence replay in Figure 5A-C using a reduced network and LTP-IE model that captures the core algorithmic properties of LTP-IE. Here we find that complex sequences can indeed be replayed by the introduction of an additional tuning parameter, such as head-direction. See Results section “Encoding complex sequences and non-spatial mappings with LTP-IE” for further details.

We have also addressed the replay of arbitrarily long sequences in the last two paragraphs of the “Model limitations**”** section of our Discussion. Specifically, we note the phenomenological connection between the gating signal used in our simulations and the theta cycles putatively thought to control sequence replay.

2) Explaining the replay of complex sequences, as mentioned above, would be a major contribution to the field, but it is not clear whether the model can generate those or not. There are many proposed models of reactivation of simple sequences, but none (as far as I know) for reactivation of complex/realistic sequences. While the authors are commended for proposing a new biophysical mechanism for explaining replay, this a moderate contribution to the field. What would be really exciting would be a model such as the one they propose that goes beyond the kind of simple sequences typically found in hippocampus studies with rodents running on linear tracks.

While we chose not to explore the replay of complex sequences in our spiking network model, we have attempted to address this in a reduced model capturing the core algorithmic properties of LTP-IE independent of its specific biophysical implementation. See our response to (1), Figure 5A-C, and Results section “Encoding complex sequences and non-spatial mappings with LTP-IE” for more details.

3) Discussion, last paragraph and elsewhere: the authors clearly state that their model works in the absence of synaptic modification, in fact this is one of the major points. This idea (that intrinsic plasticity can act based on existing, but not changing synaptic connectivity) has been presented by other groups before and this prior work should be cited and discussed in appropriate detail.For instance, previous modeling work (Salinas, 2004; Salinas, 2004) showed that changes in excitability very similar to those proposed here, in which one input modulates the response to another, are ideal for switching a network from one functional configuration (say, sensory-motor map 1) to another (say, sensory-motor map 2). Although the changes in excitability in that earlier work went by a different name, "gain modulation", the underlying mathematics would still apply, so discussing those earlier findings would bolster the argument that LTP-IE could indeed be computationally effective for the sorts of remapping proposed in this manuscript.

We thank the reviewers for pointing out this connection to gain modulation signals. Such an interaction of modulatory and input signals as described in Salinas’ 2004 papers is indeed quite conceptually and mathematically similar to the modulation of cell excitability via LTP-IE. In fact, the effective gating of input channels as described by Salinas might have additional interesting computational consequences, as it could potentially be modified even more rapidly and controllably than LTP-IE, if these signals arose from the dynamics of an upstream network. We have discussed our work in relation to Salinas’ in the fourth paragraph of our Discussion section “Comparison to existing models” and in the final paragraph of our Discussion section “Computational significance”.

The work also needs to be discussed in the context of Moshe Abeles' neuronal avalanche concept (despite its stronger focus on the neocortex).

Thank you for pointing out this connection. We have discussed a connection to Abeles’ work on synfire chains in the second paragraph of our Discussion section. We have also added a paragraph discussing the relationship between our work and neuronal avalanches, in particular the power-law distributions of spatiotemporal avalanche activity, in the final paragraph of our Discussion section “Model limitations**”**.

In the context of LTP-IE, it might be interesting to discuss a recent finding that spike firing can be independent of dendritic EPSP amplitudes, and is instead intrinsically controlled, see Ohtsuki and Hansel, 2018.

We thank the reviewers for suggesting this. We have added a reference to this idea in the fourth paragraph of our Discussion.

4) The authors state that 'this mechanism….is observed in hippocampus only'. If this statement refers to LTP-IE, it is not correct, see e.g. Daoudal, Hanada and Debanne, 2002. Debanne's work should also be cited with regard to STDP. If the statement does not refer to LTP-IE, this needs to be stated more precisely.

Thank you for pointing out this group’s work on LTP-IE, as we had not come across it before. We discuss LTP-IE outside of hippocampus in the fourth paragraph of our revised Discussion. Additionally, we have added several sentences discussing the potential interactions between LTP-IE and STDP, including the finding by Debanne’s group that the two may co-occur in physiological preparations, in the third paragraph of our Discussion section “Model limitations**”**.

5) Given the co-occurrence of replay and sharp-wave ripples, the lack of inhibition in the current model may be a substantial limitation. Replay was triggered using depolarizing current, however the physiological relevance of the model would be bolstered if replay could be triggered by a sharp-wave ripple event in an inhibitory population. More broadly, how does the model behave if one were to add inhibition in addition to excitation?

We agree that the lack of inhibition in our model was a substantial limitation and thank the reviewers for encouraging us to explore it in more depth. To this end, we have rerun all of our spiking network simulations with an inhibitory pool of neurons now included and reciprocally connected to the excitatory pool. This is reflected in the network diagram of Figure 2A and the raster and population plots in Figure 3H,I.

Adding inhibition turned out to have the beneficial effects of promoting unidirectional sequence propagation (Figure 2F, 3D) in the network and relatedly introducing a winner-take-all-like interaction between competing sequences (Figure 3J), as well as making the network more robust to parameter changes. For instance, whereas in our original excitation-only model there was only a narrow “ridge” of excitatory recurrent weights and length scales that supported activity that did not either fade out or blowup, in our model that included inhibition, the regime in which stable replay was observed without blowing up was quite large (Figure 3B). In addition, including inhibition in the network indeed yielded a sharp-wave-ripple-like oscillatory component to the spontaneously generated replay events, depicted in a power spectrum analysis of the excitatory population activity (Figure 3H). We note, however, that in our model the spontaneous replay/sharp-wave-ripple events arise initially in the excitatory network, which triggers the back-and-forth oscillatory interaction with the inhibitory network. This appears to be consistent with some research that finds SWR origination to occur in CA3 and to subsequently propagate elsewhere [Buzsaki 1986; Csicsvari 2000; Carr 2011; Ramirez-Villegas 2018].

6) Related to the above point, it was difficult to evaluate the robustness of the model. There are many parameters, but which ones are more critical for the results, and what is the 'operating range' of the model? Is the model robust to small perturbations in parameters?

While a full parameter search was prohibitively computationally expensive, we have explored the dependence of replay occurrence on a handful of pairs of parameters we believed to be crucial to the network dynamics, including the strength of excitation and inhibition, as well as the gating input rate and maximum LTP-IE level. These are shown in Figure 3A-C. The observed robustness of our E-I spiking network to variation in excitatory connection strength also suggests that the network may be able to contain heterogeneous network structures encoding long-term memories while still supporting short-term LTP-IE-dependent sequence replay; we have discussed this notion in the third paragraph of our Discussion section “Comparison to existing models”. We have also explored the parameter dependence of “virtual replay speed” in Figure 3F-G.

7) Regarding Figure 4, it is unclear where in the brain a recurrent sensorimotor network with LTP-IE might exist to support the generation of transient associations using the proposed mechanism. What do the authors suggest is the biophysical substrate for non-spatial transient associations?

Thank you for pointing out this aspect of our original results. While we had briefly discussed LTP-IE in non-spatial context in our original Discussion, we agree that a more thorough investigation of this idea was worthwhile to pursue. To this end we have used our reduced model not only to investigate complex/intersecting trajectories but also to demonstrate the algorithmic potential of LTP-IE to encode stimulus-response relationships without a spatially organized network. Specifically, we have simulated a network similar to Swan and Wyble’s “binding pool” network [Swan, 2014], except using slightly different neural dynamics as well as multi-neuron ensembles to represent the “items” to be bound together, and importantly using LTP-IE as the memory substrate, rather than persistent activity. Our results are shown in Figure 5D-E and discussed in Results section “Encoding complex sequences and non-spatial mappings with LTP-IE” and in an additional couple of sentences in the second paragraph of our Discussion section **“**Computational significance**”**.

8) Some parts of the manuscript were found to be somewhat technical and difficult to follow. Specifically, throughout the Results: second paragraph; subsection “LTPIE increases membrane voltages and spike rates under random gating inputs”, first and last paragraphs; subsection “Spike sequences propagate along LTP-IE-defined paths through a network”, second paragraph; subsection “Dependence of LTP-IE-based sequence propagation on network parameters”, first paragraph and subsection “LTP-IE-based sequences can encode temporary stimulus response mappings”, last paragraph. The Results should be more accessible to a broader readership not specialized in computational neuroscience.For instance, it is not easy to understand how the results in Figure 2 relate to, and emerge from, those in Figure 1. In general, it would be useful if the authors could better explain the relationship between their neuronal mechanism and the results.

Thank you for pointing out lines that were not clear upon reading. We have rewritten these specified sentences in a way that we hope makes them more transparent to an audience not specialized in computational neuroscience.

Additionally, to clarify the connection between Figure 1 and Figure 2 we have added a couple of new clarifying sentences in Results section “Spike sequences propagate along LTP-IE-defined paths through a network”.

In addition, while the model works well for encoding spatial trajectories in which the closeness of individual locations is pre-encoded in the recurrent connectivity weights, it is difficult to see how this would extend to the given example of cognitive flexibility (i.e., raising our right hand when a particular word is heard, and the left hand when a different word is heard), particularly if no a priori spatial relationship exists. Are spatial relationships necessary? There is no evidence provided to suggest that the neocortical network has the necessary architecture to support LTP-IE based replay.

Thank you for raising these questions. While our spiking network model includes spatially organized connectivity, we do not believe spatial organization is strictly necessary for LTP-IE-dependent mappings. Specifically, we have addressed cognitive flexibility and transient stimulus-response mappings in networks with no a priori spatial relationship using our reduced model. See Figure 5D-E and our response to (7).